# FADD is recruited to activated STING oligomers to initiate caspase-mediated NF-κB activation in *Drosophila melanogaster*

Kasper Grønbjerg Winther[1,2,5], Juliette Schneider[1,5], Gabrielle Haas[1], Anna Hvarregaard Christensen[2],
Shreya Goyal[1], Weisheng Luo [3], Ziming Wei [3], Jiyong Liu[3], Katarzyna Kjøge [2], Jan J Enghild [2],
Carine Meignin [1], Neal Silverman [4], Hua Cai[3], Jean-Luc Imler [1✉] & Rune Hartmann [2✉]

## Abstract

**STING is an evolutionarily conserved key regulator of innate immunity. In the model organism *Drosophila melanogaster*, STING activates the NF-κB-like transcription factor Relish, initially characterized for its role in the antibacterial IMD pathway. The versatile FADD/Caspase-8 axis is widely used in various immune signaling pathways throughout the animal kingdom, including the IMD pathway. Here, we show that it functions downstream of STING in *Drosophila* to mediate Relish activation by the Caspase-8 homolog DREDD. We present a detailed structural model illustrating how the adapter protein FADD interacts with two separate STING dimers in the activated oligomerized form of STING, thus providing a molecular explanation for the activation-dependent recruitment of FADD. We further show that FADD interacts with IMD in a structurally distinct but functionally related manner, highlighting how the STING and IMD pathways differentially utilize the adapter protein FADD. Our results illustrate how an ancestral module is incorporated into different innate immune pathways, providing insights into the evolution of host-pathogen interactions.**

**Keywords** Innate Immunity; FADD; STING; Death Fold Domain; *Drosophila Melanogaster*

**Subject Categories** Immunology; Microbiology, Virology & Host Pathogen Interaction

## Introduction

Viruses pose a fundamental threat to all living cells. To resist this threat, animals rely on a set of sophisticated antiviral immune mechanisms. Some of these mechanisms are evolutionarily ancient, and their origin can be traced back to prokaryotes (Jenson et al, 2023; Morehouse et al, 2020; Ofir and Sorek, 2018; Rousset et al, 2023). Indeed, recent results revealed that a set of modules (proteins or domains) operating in innate immune pathways of animals or plants originates from antiphage modules identified in prokaryotic immune defenses (Bernheim et al, 2024; Wein and Sorek, 2022). In the course of evolution, these antiphage modules have been acquired by eukaryotes through vertical or horizontal gene transfer and recombined to build tailored innate immunity pathways. One such ancestral antiviral defense mechanism is the STING pathway, which is broadly conserved among multicellular animals (Barber, 2014; Slavik and Kranzusch, 2023). In humans, the STING pathway is activated by the immune sensor cyclic GMP-AMP synthase (cGAS), that detects dsDNA in the cytosol and generates a second messenger, cyclic 2'-3'-GMP-AMP (cGAMP), which in turn binds and activates STING (Ablasser et al, 2013; Civril et al, 2013; Diner et al, 2013; Gao et al, 2013; Sun et al, 2013; Wu et al, 2013). Signaling downstream of STING results in induction of a broad antiviral and inflammatory transcriptional response, as well as autophagy activation (Zhang et al, 2019). In bacteria, cGAS-like enzymes are activated upon sensing of phage infection and trigger the production of cyclic di- or tri-nucleotides (CDNs, CTNs), which bind to and activate effector antiphage molecules. These effectors include, although they are not limited to, STING homologs (Morehouse et al, 2020; Slavik and Kranzusch, 2023).

We and others identified two active cyclic nucleotidyltransferase enzymes with homology to human cGAS in *Drosophila melanogaster*, and named them cGAS-like receptor (cGLR) 1 and 2. Both those enzymes are required in a partially redundant manner for flies to mount an efficient antiviral response against both RNA and DNA viruses (Cai et al, 2023; Holleufer et al, 2021; Slavik et al, 2021). In mammals, CDNs produced by cGAS bind to STING and initiate a transcriptional response via the transcription factors interferon regulatory factor 3 (IRF3) and nuclear factor kappa-light-chain-enhancer of activated B cells (NF-κB) (Ishikawa and Barber, 2008; Liu et al, 2015). While molecular and structural studies have provided a reasonable understanding of the

[1]University of Strasbourg, CNRS UPR9022, Strasbourg, France. [2]Department of Molecular Biology and Genetics, Aarhus University, Aarhus, Denmark. [3]Sino-French Hoffman institute, School of Basic Medical Science, Guangzhou Medical University, Guangzhou, China. [4]Program in Innate Immunity and Division of Infectious Diseases and Immunology, Department of Medicine, University of Massachusetts Chan Medical School, Worcester, MA, USA. [5]These authors contributed equally: Kasper Grønbjerg Winther, Juliette Schneider. ✉E-mail: jl.imler@unistra.fr; rh@mbg.au.dk

mechanism leading to activation of IRF3 by the kinase TANK-binding kinase 1 (TBK1) downstream of STING (Ergun et al, 2019; Zhang et al, 2019; Zhao et al, 2019), activation of NF-κB remains poorly understood. Activation of those two pathways lead to expression of both antiviral proteins, including interferons (IFNs), and proinflammatory cytokines. Gain-of-function variants of STING are found in human patients suffering from STING-associated vasculopathy with onset in infancy (SAVI) (Jeremiah et al, 2014; Liu et al, 2014). Expression of those variants in mice drives the induction of inflammatory disease in an IRF3-independent manner (Bouis et al, 2019; Gao et al, 2024; Warner et al, 2017), suggesting that NF-κB is a key driver of STING-mediated inflammation. *Drosophila*, like other insects, does not appear to encode any IRF-like transcription factors and activation of *Drosophila* STING leads to a transcriptional response mediated by the NF-κB factor Relish, which initiates transcription of a set of antiviral genes in flies (Cai et al, 2020; Goto et al, 2018; Holleufer et al, 2021; Hua et al, 2018; Martin et al, 2018; Slavik et al, 2021). This and independent work in the sea anemone *Nematostella vectensis* revealed that activation of the NF-κB pathway is an ancestral feature of STING signaling, raising the question of the mechanism involved (Kranzusch et al, 2015; Margolis et al, 2021).

In *Drosophila melanogaster*, the NF-κB-like transcription factor Relish was initially characterized for its role in the immune deficiency (IMD) pathway, which is critical for antibacterial immunity (Hultmark, 2003; Lemaitre and Hoffmann, 2007). Recognition of di-amino pimelic (DAP) type peptidoglycan (PGN), a conserved bacterial cell wall component, by either the transmembrane peptidoglycan recognition protein LC (PGRP-LC) or the intracellular PGRP-LE leads to activation of Relish via the death domain (DD) adapter protein IMD (Royet and Dziarski, 2007). IMD signaling involves the formation of amyloid fibrils and the recruitment of the adapter molecule *Drosophila* FADD (dFADD), a homolog of human Fas-associated via death domain (FADD), as well as Death-related ced-3/Nedd2-like caspase (DREDD), a homolog of human Caspase 8 (Kleino et al, 2017; Leulier et al, 2000; Naitza et al, 2002). DREDD cleaves Relish at aspartate 545 (D545), releasing the N-terminal Rel-homology domain (RHD) from the C-terminal inhibitory ankyrin repeat domain (Stoven et al, 2003). DREDD also cleaves the N-terminus of IMD, revealing an inhibitor of apoptosis binding motif that mediates association with the E3 ubiquitin ligase *Drosophila* inhibitor of apoptosis 2 (DIAP2), leading to rapid conjugation of IMD with K63-linked polyubiquitin chains (Paquette et al, 2010). This allows for the recruitment of the kinases *Drosophila* TGF-β activated kinase 1 (dTAK1) and *Drosophila* IκB kinase β (dIKKβ, encoded by the gene *ird5* in *Drosophila*) together with the regulatory subunits dTAK1-associated binding protein 2 (dTAB2) and *Drosophila* IKKγ (dIKKγ, encoded by the gene *key* in *Drosophila*), resulting in phosphorylation of Relish. Following these events, the N-terminal RHD-containing fragment of Relish translocates to the nucleus where it induces expression of antimicrobial peptides (AMPs), such as cecropins, attacins or diptericins (Ganesan et al, 2011).

By contrast, it is still unclear how Relish is activated by *Drosophila* STING (dSTING). Indeed, our initial characterization of the dSTING pathway only pointed to a contribution of the IKKβ kinase (Goto et al, 2018). In addition, other studies reported that dSTING functions in the IMD-Relish axis to induce expression of AMPs in response to infection by the bacteria *Listeria monocytogenes* and *Coxiella burnetii*, casting doubts on the existence of two independent pathways regulating activation of Relish (Guzman et al, 2024; Martin et al, 2018). Here, we show that dFADD is specifically recruited to activated, oligomeric dSTING to trigger cleavage and activation of Relish in a DREDD-dependent but IMD-independent manner. We provide a detailed three-dimensional structural model supported by mutational analysis of how dFADD interacts with two separate dSTING dimers in the context of an activated dSTING oligomer, thus ensuring that dFADD only interacts with the active form of dSTING. Finally, we show that both dFADD and DREDD are required for the antiviral activity of dSTING in vivo.

## Results

### dSTING activation triggers cleavage of Relish

A key aspect of NF-κB signaling is the signal-dependent degradation of inhibitor of NF-κB proteins (IκBs), which liberates the RHD for nuclear translocation (Silverman et al, 2000; Zhang et al, 2017). Relish is a class I NF-κB transcription factor, which includes a C-terminal ankyrin repeat domain similar to IκBs in addition to the N-terminal RHD-containing transcription factor module. Like its mammalian homologs, NF-κB1 and NF-κB2, the inhibitory domain needs to be released before Relish can activate transcription (Kim et al, 2014; Stoven et al, 2000). We and others previously demonstrated a genetic requirement of Relish in transcriptional induction downstream of dSTING, although it is not known if Relish is actively controlled by dSTING (Goto et al, 2018; Hua et al, 2018). To determine if dSTING activation directly leads to cleavage of Relish, we expressed Relish with a C-terminal V5-tag in *D. melanogaster* S2 cells and activated dSTING or IMD signaling (positive control) by expression of the receptors cGLR1 or PGRP-LC, respectively. We detected a clear increase in cleavage of Relish upon both IMD and dSTING activation (Fig. 1A). Mutation of the DREDD cleavage site found within Relish (D545A) completely blocked cleavage induced by both pathways.

### Relish cleavage is required for dSTING signaling

Having established that dSTING activation leads to Relish cleavage, we asked if this is required for transcriptional induction of target genes, and therefore we generated *Relish* knock-out (KO) S2 cell pools (Appendix Fig. S1). We used an *Attacin A (AttA)* luciferase reporter to measure IMD activation. Expression of PGRP-LC in wild-type (WT) S2 cells led to activation of this reporter and cleavage of Relish (Fig. EV1A,B). Activation of the *AttA* reporter was prevented upon disruption of the *Relish* gene but restored upon ectopic expression of Relish. Similar results were seen using cGLR1 to activate a *Sting* reporter (Fig. EV1C,D), placing Relish downstream of both dSTING and IMD signaling. Interestingly, expression of the non-cleavable form of Relish (D545A) failed to rescue signaling in the *Relish* deficient cells and acted as a dominant negative mutation in the WT cells for both reporters. Finally, expression of a C-terminally truncated form of Relish (1–545) corresponding to the cleaved form of Relish, led to activation of both reporters independently of the presence of cGLR1 or PGRP-LC (Fig. EV1A,C).

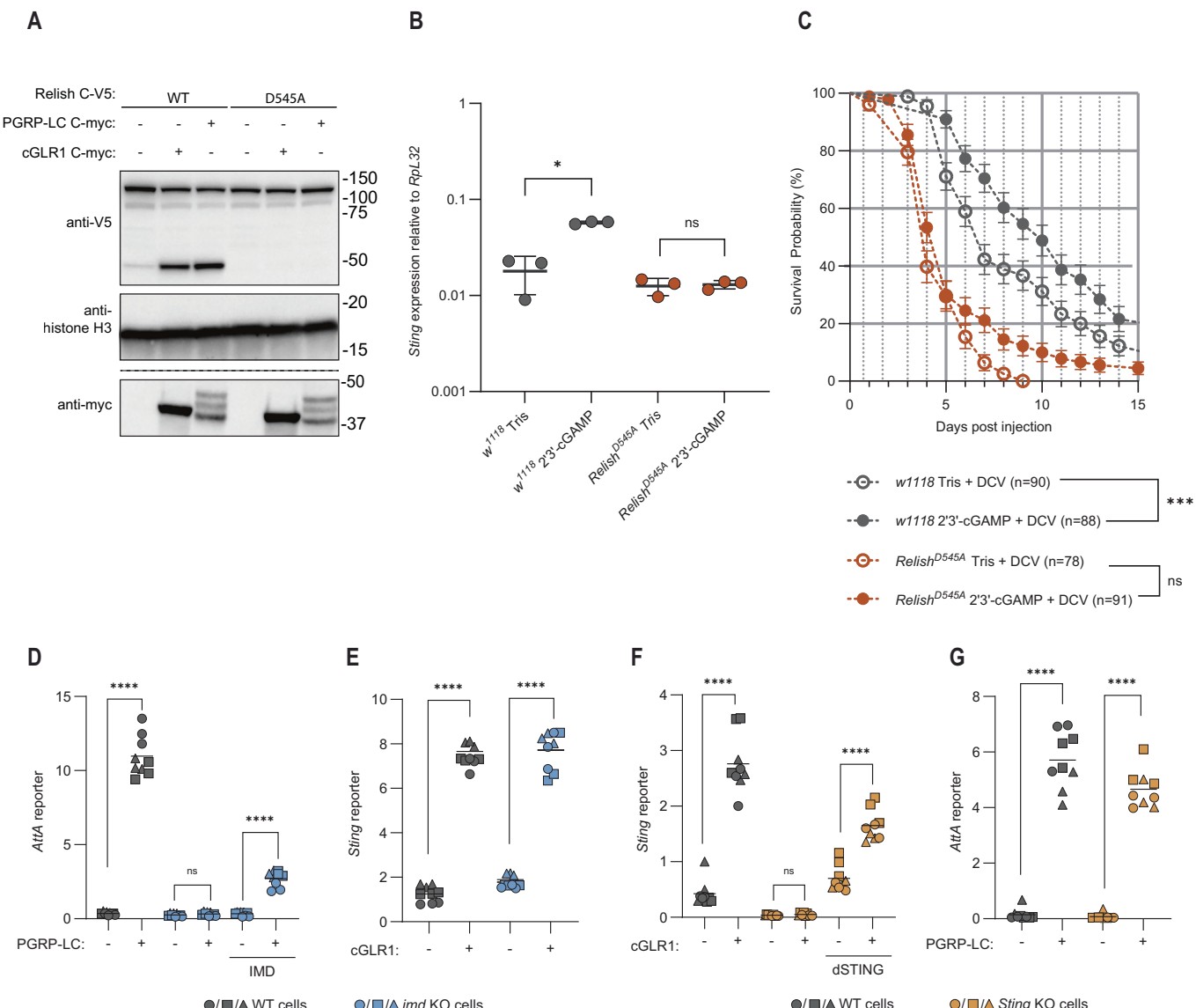

**Figure 1. dSTING signaling leads to cleavage of the NF-κB transcription factor Relish and is independent of IMD signaling.**

(A) Cleavage of ectopically expressed WT or D545A mutant Relish C-V5 in S2 cells in response to dSTING activation (cGLR1 expression) or IMD activation (PGRP-LC expression). Representative of $n = 3$ independent experiments. (B) Induction of *Sting* in $w^{1118}$ (control) or $Relish^{D545A}$ flies measured by qPCR 24 h after intrathoracic injection of the STING agonist 2'3'-cGAMP. Each data point is derived from a pool of six flies (three male, three female). Bars represent mean ± standard deviation. Logarithmic scale. *p* values were calculated using a pairwise permutation test corrected with the Benjamini–Hochberg method: $*p = 0.03866$ and ns: $p = 0.7745$. (C) Survival of control $w^{1118}$ (control) or $Relish^{D545A}$ flies injected with DCV and co-injected with Tris or 2'3'-cGAMP. Points and bars represent mean ± standard error. *P* values were calculated with a Gehan–Breslow–Wilcoxon test: $***p = 0.0004$ and ns: $p = 0.1025$. (D, E) Induction of *AttA* luciferase reporter upon expression of PGRP-LC (D) or *Sting* luciferase reporter upon expression of cGLR1 (E) in WT (gray) or *imd* KO (blue) S2 cells. Cells were complemented with IMD as indicated. (F, G) Induction of the *Sting* luciferase reporter upon expression of cGLR1 (F) or *AttA* luciferase reporter upon expression of PGRP-LC (G) in WT (gray) or *Sting* KO (yellow) S2 cells. Cells were reconstituted with dSTING as indicated. In D–G data from three independent experiments (different geometrical icons), each performed in biological triplicates, are shown with means ($n = 9$). *p* values were calculated using two-way analysis of variance (ANOVA) corrected with a two-tailed Holm–Šídák post hoc test. $****P < 0.0001$, ns: $P = 0.9949$ (D) or $P > 0.9999$ (F). Source data are available online for this figure.

To test the requirement for Relish cleavage in the dSTING-induced transcriptional response in vivo, the STING agonist 2'3'-cGAMP was injected intrathoracically into either control flies or knock-in flies harboring the D545A mutation at the endogenous *Relish* locus. The injection induced transcription of *Sting*-regulated genes (*Srgs*), including *Sting* itself, in control flies (activation of dSTING induces its own expression)(Goto et al, 2018), whereas $Relish^{D545A}$ mutant flies

did not respond (Figs. 1B and EV1E–G). Furthermore, $Relish^{D545A}$ mutant flies succumbed more rapidly than controls when challenged with the viral pathogen *Drosophila* C virus (DCV). Note that $Relish^{D545A}$ mutant flies did also show a reduced lifespan upon injection of Tris buffer, suggesting some generalized frailty in this strain possibly due to environmental microbes. Injection of 2'3'-cGAMP protected control flies, but not $Relish^{D545A}$ mutant flies, against DCV infection

(Figs. 1C and EV1H). Altogether, these results support the hypothesis that activation of Relish by proteolytic cleavage is required for dSTING-mediated transcriptional induction and the associated antiviral immunity.

## dSTING and IMD are independent activators of Relish

dSTING and IMD represent two structurally distinct activators of innate immune pathways that both utilize Relish to drive their transcriptional response. However, conflicting data question their interdependence and their capacity to establish separate signalosomes within the cell upon activation (Guzman et al, 2024; Martin et al, 2018; Aalto et al, 2023). Therefore, we tested if the dSTING and IMD proteins signal independently of each other in S2 cells. Disrupting the *imd* gene prevented PGRP-LC-mediated activation of the *AttA* reporter, unless rescued by ectopic expression of IMD. By contrast, cGLR1-mediated activation of the *Sting* reporter was largely unaffected in *imd* deficient cells (Fig. 1D,E; Appendix Fig. S1). Inversely, disrupting the *Sting* gene prevented cGLR1-mediated activation of the *Sting* reporter, whilst it did not affect PGRP-LC-mediated activation of the *AttA* reporter (Fig. 1F,G; Appendix Fig. S1).

The kinase dTAK1 and its regulatory subunit dTAB2 have an established role downstream of IMD signaling (Kleino et al, 2005; Zhuang et al, 2006). To investigate their potential role in dSTING signaling, we tested the in vivo requirement of dTAK1 for dSTING. The *Tak1* deficient flies responded to injection with 3'2'-cGAMP - a more potent dSTING agonist than 2'3'-cGAMP in flies - in a similar manner to control flies (Fig. EV2A–C). However, their response to a challenge with Gram-negative bacteria was reduced, as expected (Fig. EV2D,E). We confirmed these results in *Tak1* and *Tab2* KO S2 cells. A clear reduction of IMD signaling was observed in both *Tak1* and *Tab2* deficient cells, whereas dSTING signaling was unaffected. Thus, dSTING signaling does not rely upon the dTAK1/dTAB2 complex (Fig. EV2F,G).

Altogether, these data indicate that, although both IMD and dSTING trigger cleavage of Relish at D545, the two pathways are activated by different receptors and signal independently of each other at the level of the IMD and dSTING proteins.

## DREDD and dFADD are required for dSTING-mediated cleavage of Relish and antiviral immunity

The caspase DREDD is known to cleave Relish in response to IMD signaling (Stoven et al, 2000). Therefore, we tested if DREDD is required for dSTING-mediated cleavage of Relish at D545. For this, we generated S2 cells lacking a functional *Dredd* gene (Appendix Fig. S1). These cells failed to cleave Relish in response to either cGLR1 or PGRP-LC stimulation. However, this could be rescued by ectopic expression of WT DREDD, but not the catalytically inactive C386A mutant (Fig. 2A). Expression of both WT and mutant DREDD was verified by western blot (Fig. EV3A). Deletion of the *Dredd* gene in flies similarly led to a loss of gene induction in response to injection of 3'2'-cGAMP compared to control flies (Figs. 2B and EV3B–D). Furthermore, *Dredd* mutant flies exhibited a reduced lifespan upon DCV infection, compared to control flies and were not protected by 3'2'-cGAMP injection (Fig. 2C). Thus, DREDD is needed for dSTING-mediated cleavage of Relish and hence for the antiviral activity of dSTING, raising the question of how DREDD is recruited to the dSTING signalosome.

In the IMD pathway, DREDD is recruited by the adapter dFADD, which binds to IMD via interactions between the death domains of these two proteins. In turn, dFADD and DREDD interact via their death effector domains. STING does not possess a death domain nor a death effector domain in any metazoan species. Therefore, we tested whether dFADD participates in dSTING signaling using a similar approach as described above. Accordingly, disruption of the *Fadd* gene in S2 cells (Appendix Fig. S1) prevented both dSTING- and IMD-mediated cleavage of Relish, and this could be rescued by exogenous expression of dFADD (Fig. 2D). Likewise, mutation of the *Fadd* gene in flies led to a loss of induced *Sting* and *Srg2* expression upon 3'2'-cGAMP injection compared to control flies. Induction was not abolished for *Srg1* and *Srg3*, although their basal level of expression was significantly reduced (Figs. 2E and EV3E–G). Furthermore, *Fadd* mutant flies exhibited a reduction of lifespan upon DCV infection compared to WT flies and the protective effect of 3'2'-cGAMP injection was abrogated in these flies (Fig. 2F). Altogether, our data reveals that both DREDD and dFADD, along with Relish, are required for the antiviral activity of dSTING.

## dFADD interacts directly with the activated oligomeric form of dSTING

Our findings raise the question of how dSTING recruits dFADD and DREDD? Prior to activation, STING exists as a dimeric molecule and CDN binding triggers oligomerization, a key characteristic of STING activation (Saitoh et al, 2009; Shang et al, 2019). To test the hypothesis that dSTING and dFADD interact, we made use of AlphaFold, which has a unique ability to predict protein-protein interactions (Jumper et al, 2021). Despite several attempts, we were unable to predict any plausible interactions between dFADD and the dimeric form of dSTING in silico. However, when we allowed for the formation of tetrameric dSTING (representing the first step in oligomerization), we were able to predict a possible interaction. This prediction suggests that the death domain of dFADD binds directly to the interface between two dSTING dimers (Fig. 3A; Appendix Fig. S2). This model provides an explanation for the selective affinity of dFADD for activated dSTING, since only oligomeric and thereby activated dSTING will be able to recruit dFADD. According to our model, R164 in dFADD forms strong interactions with E183 and D184 in one dSTING dimer. Simultaneously, D181 and E184 in dFADD interact with R240 in the adjacent STING dimer (Fig. 3A). To confirm our model, we designed a series of mutations of both dFADD and dSTING based upon this model. These mutations were tested for their ability to induce dSTING signaling in *Fadd-* or *Sting*-deficient S2 cells, respectively. Individual mutation of R164 or D181 to alanine (R164A and D181A) in dFADD did not have a significant effect compared to WT dFADD on *Sting* reporter activity in *Fadd*-deficient cells. However, combining these mutations resulted in strong attenuation of reporter signaling (Figs. 3B and EV4A). Mutation of both E183 and D184 to alanine (E183A and D184A) in dSTING (which are predicted to interact with R164 in dFADD) resulted in a modest yet significant reduction in *Sting* reporter activity in *Sting*-deficient cells compared to WT dSTING, whereas mutation of R240 (which is predicted to interact with D181 in dFADD) to alanine (R240A) had a stronger impact. Furthermore, combining these mutations (E183A/D184A/R240A)

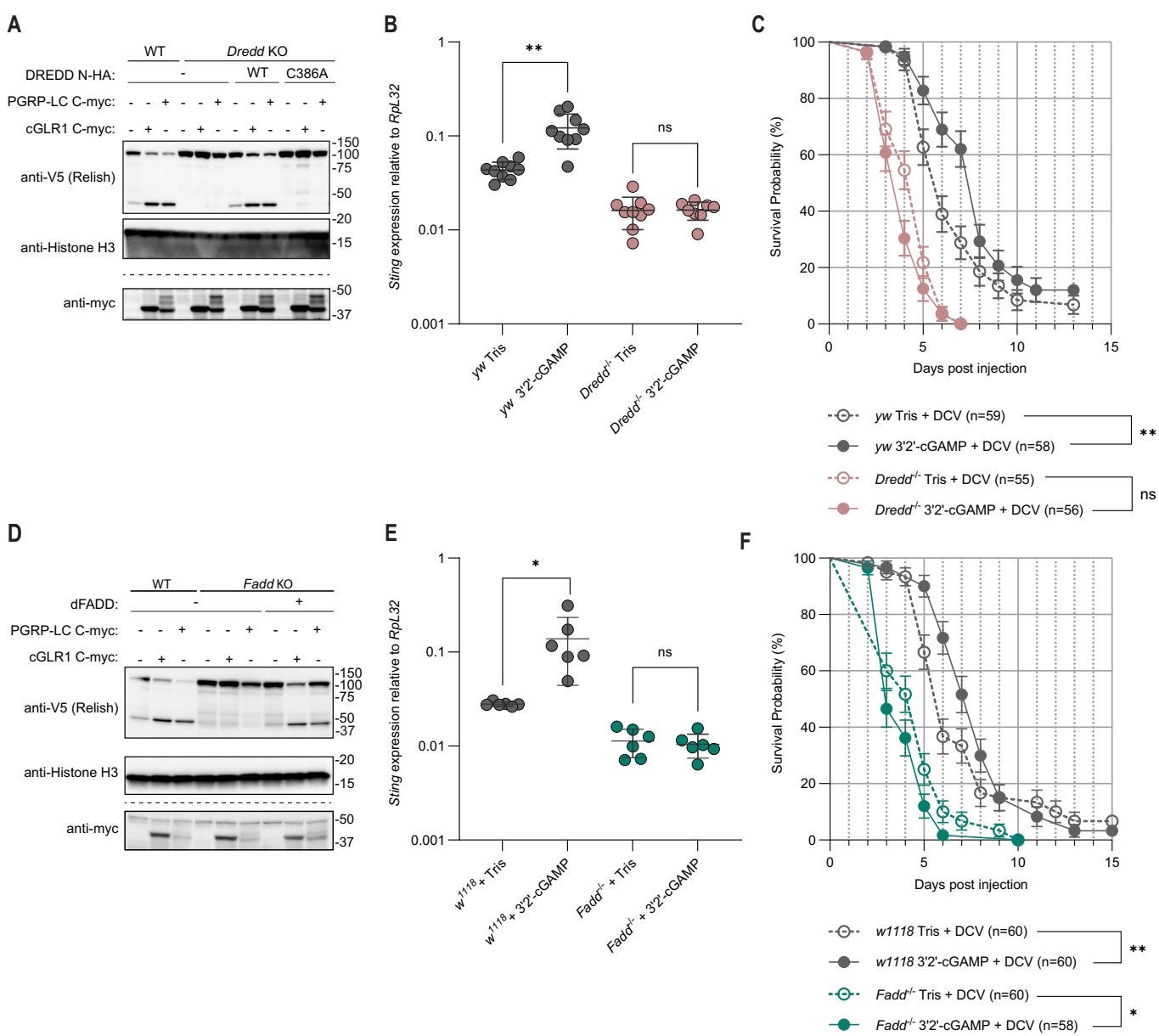

**Figure 2. The dFADD:DREDD complex activates Relish downstream of dSTING.**

(A) Cleavage of ectopically expressed Relish C-V5 in WT and *Dredd* KO S2 cells in response to expression of cGLR1 or PGRP-LC. *Dredd* KO cells were reconstituted with either WT DREDD N-HA or a catalytically inactive mutant (C386A) as indicated. Representative of $n = 2$ independent experiments. (B) Induction of *Sting* in *yw* (control) or *yw*$^{Dredd-/-}$ flies measured by qPCR 24 h after intrathoracic injection of the STING agonist 3'2'-cGAMP or Tris control. Each data point is derived from a pool of six flies (three male, three female). Bars represent mean ± standard deviation. *p* values were calculated using a pairwise permutation test corrected with the Benjamini–Hochberg method. **$p = 0.002458$ and ns: $p = 0.3448$. (C) Survival of *yw* (control) or *yw*$^{Dredd-/-}$ flies injected with DCV and co-injected with 3'2'-cGAMP or Tris control. Points and bars represent mean ± standard error. *p* values were calculated with a Gehan–Breslow–Wilcoxon test: **$p = 0.0026$ and ns: $p = 0.0647$. (D) Cleavage of ectopically expressed Relish C-V5 in WT and *Fadd* KO S2 cells in response to expression of cGLR1 or PGRP-LC. *Fadd* KO cells were reconstituted with WT dFADD as indicated. Representative of $n = 3$ independent experiments. (E) Induction of *Sting* in *w*$^{1118}$ (control) or *Fadd*$^{-/-}$ flies measured by qPCR 24 h after intrathoracic injection of 3'2'-cGAMP or Tris control. Each data point is derived from a pool of six flies (three male, three female). Bars represent mean ± standard deviation. *p* values were calculated using a pairwise permutation test corrected with the Benjamini–Hochberg method. *$p = 0.02955$ and ns: $p = 0.6531$. (F) Survival of *w*$^{1118}$ (control) or *Fadd*$^{-/-}$ flies injected with DCV and co-injected with 3'2'-cGAMP or Tris control. Points and bars represent mean ± standard error. *p* values were calculated with a Gehan–Breslow–Wilcoxon test: *$p = 0.0401$ and **$p = 0.0024$. Source data are available online for this figure.

led to an almost complete loss of *Sting* reporter signaling (Figs. 3C and EV4B).

To confirm a physical interaction between dFADD and activated dSTING, we transfected S2 cells with a tagged version of dSTING and used this tag to immunoprecipitate dSTING. Subsequently, we

used liquid chromatography—mass spectrometry (LC/MS) to detect peptides derived from dFADD. In the absence of activation, the quantity of dFADD-derived peptides was comparable to the background level (EGFP). Yet, when dSTING was activated by co-transfection with cGAS, we observed a strong increase in the

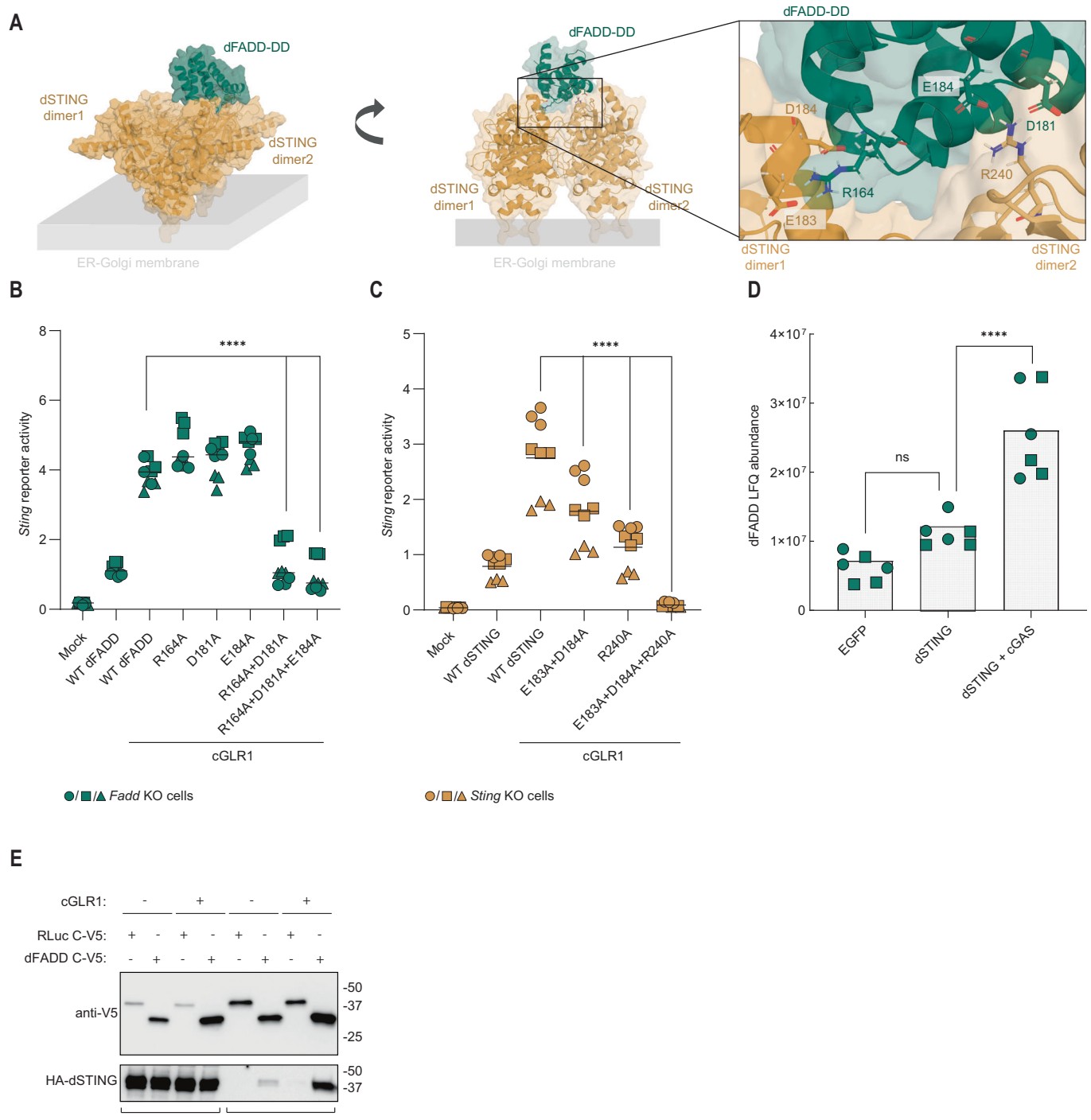

**Figure 3. dFADD interacts with the oligomerized form of dSTING.**

(A) In silico prediction of the dFADD death domain (DD) (green) binding to the interface of oligomerized dSTING dimers (yellow). The insert shows key interacting residues. (B) Induction of *Sting* reporter in *Fadd* KO S2 cells upon expression of cGLR1 and reconstitution with WT dFADD or mutants disrupting the predicted dSTING:dFADD interface. (C) Induction of the *Sting* reporter in *Sting* KO S2 cells upon co-expression of cGLR1 and reconstitution with WT dSTING or mutants disrupting the predicted dSTING:dFADD interface. (D) LC/MS -based detection of dFADD peptides from co-immunoprecipitation of ectopically expressed EGFP (negative control) or V5-tagged dSTING in WT S2 cells with or without co-expression of cGAS. Data from $n = 2$ independent experiments, each containing three biological replicates, are shown. Bars indicate the mean. *p* values were calculated using two-way ANOVA, corrected with a two-tailed Holm–Šídák post hoc test: ****$p < 0.0001$, ns $= 0.1167$. (E) Co-immunoprecipitation of dSTING N-HA and dFADD C-V5 ectopically expressed in WT S2 cells with or without co-expression of cGLR1. Representative of $n = 2$ independent experiments. In **B**, **C** data from three independent experiments (different geometrical icons), each performed in biological triplicate, are shown with mean ($n = 9$). *p* values were calculated using two-way ANOVA, corrected with Dunnett's post hoc test: ****$p < 0.0001$. Source data are available online for this figure.

quantity of dFADD-derived peptides that co-precipitated with dSTING (Fig. 3D), yet the level of dSTING did not change (Fig. EV4F). Furthermore, DREDD peptides were also detected in the condition with activated dSTING, but at borderline significance, as this is a secondary interaction to dSTING (Fig. EV4G). Classical co-immunoprecipitation confirmed an interaction between dFADD and dSTING. Furthermore, the ability of dFADD to co-immunoprecipitate dSTING was substantially increased upon activation of dSTING by cGLR1 co-expression (Fig. 3E). Thus, the mutational analysis supports our AlphaFold prediction, and an activity-dependent direct interaction was verified by immunoprecipitation.

Overall, our data support the hypothesis that two separate sites within the death domain of dFADD interact with two separate dSTING dimers bridged together in the activated oligomer of dSTING. This provides an explanation for the specific recruitment of dFADD to activated dSTING, and we propose that recruitment of dFADD is one of the critical first steps in forming a functional dSTING signalosome.

## dFADD and DREDD interact via a classical death effector domain interaction

dFADD binds DREDD via its N-terminal death effector domain (DED), which interacts with the cognate domain in DREDD (Hu and Yang, 2000; Leulier et al, 2002) and this axis is conserved throughout the animal kingdom. We hypothesized that the function of dFADD in dSTING signaling is to recruit DREDD to the dSTING signalosome and to position it so that it can cleave Relish. To visualize the dFADD:DREDD interaction, we again used AlphaFold to predict a structural model of the complex (Fig. 4A; Appendix Fig. S3). Our model predicts an interaction of the positively charged residues R39 and R40 in dFADD with the negatively charged residues E131 and E167 in DREDD. To verify this model, we first introduced the R39A/R40A double mutation into dFADD. This mutation abolished the ability of dFADD to mediate both dSTING and IMD signaling in Fadd- deficient cells (Figs. 4B,C and EV4C,D) as well as the interaction between dFADD and DREDD, as shown by classical co-immunoprecipitation (Fig. 4D). Next, we introduced the E131A or E167A mutations as well as the double mutation E131A/E167A in DREDD. E131A alone did not have a significant effect on either Sting or AttA reporter activity in Dredd-deficient cells, while E167A led to a clear decrease in activity for both reporters. The double mutant led to a complete loss of activity for both dSTING- and IMD-induced signaling (Fig. 4E,F and EV4E) and abolished the interaction between dFADD and DREDD, as shown by classical co-immunoprecipitation (Fig. 4G). Thus, our mutational analysis and co-immunoprecipitation assays verify the AlphaFold model and confirm that dFADD and DREDD interact through a classical death effector domain interaction. Taken together, our data show that dFADD recruits DREDD to the two independent signaling complexes created by either IMD or dSTING, so that it can cleave Relish.

## dFADD interacts with IMD via its death domain

We used AlphaFold to predict an interaction between the death domains of IMD and dFADD. In the predicted complex, R171 in dFADD forms strong interactions with E220 and Y223 in IMD.

This model also suggests that E177 in dFADD is indirectly important for the interaction by stabilizing the conformation of R171 and thus reducing the entropic cost of binding (Fig. 5A; Appendix Fig. S4). To confirm the model, we mutated the residues E220 or Y223 in IMD to alanine (E220A and Y223A), which abolished AttA reporter activity in imd deficient cells, showing defective IMD signaling (Figs. 5B and EV5A). Next, we immuno-precipitated a tagged version of IMD and tested for co-immunoprecipitation of dFADD. Indeed, the interaction was lost upon mutation of either E220 or Y223 (Fig. 5C).

Similarly, we found that mutation of E177 in dFADD to alanine (E177A) led to a severe impairment of AttA reporter activity in Fadd- deficient cells, while mutation of R171 to alanine (R171A) completely abolished AttA reporter activation (Figs. 5D and EV4D). Additionally, co-immunoprecipitation experiments showed that the E177A mutation in dFADD led to a strong decrease in co-immunoprecipitated IMD, whereas the R171A mutation led to a complete loss of this interaction (Fig. 5E). Altogether, these data indicate that the residues E220 and Y223 in IMD interact with R171 in dFADD, with a contribution of E177.

Interestingly, while it is the same general area of dFADD that interacts with either dSTING or IMD, according to our AlphaFold model, the residues R171 and E177 in dFADD only form weak non-ionic polar bonds with dSTING. Indeed, introduction of the R171A and E177A mutations in dFADD had no detectable effect on the ability of dFADD to mediate dSTING signaling in Fadd-deficient cells (Figs. 5F and EV4C). However, the same was not true for R164A/D181A and R164A/D181A/E184A mutations of dFADD, which were designed to abolish dSTING activation. The double mutation R164A/D181A almost abolished IMD signaling, whereas the triple mutation R164A/D181A/E184A had a more intermediate phenotype (Fig. EV5B). Since it is the same area of dFADD which is responsible for interacting with either dSTING or IMD, it is not surprising that the mutations designed to abolish the binding to dSTING also have some effect on the interaction with IMD. We cannot offer an explanation to why the triple mutation is less severe than the double mutation. As controls, we also tested if the substitutions R171A and E177A affected the ability of dFADD to bind DREDD, which they did not (Fig. EV5C). Similarly, mutation of the DREDD binding motif in dFADD (R39A/R40A) did not affect binding of the protein to IMD (Fig. EV5D).

## Discussion

### IMD and STING represent two independent pathways, both using FADD and DREDD to cleave Relish

Early reports connecting the IMD pathway to antiviral immunity, along with the fact that dSTING activates the same NF-κB factor, Relish, as IMD signaling initially made us question whether dSTING and IMD acted in the same or different pathways (Avadhanula et al, 2009; Costa et al, 2009). Indeed, others have suggested that dSTING acts downstream of IMD or at least depends upon IMD for its activity (Guzman et al, 2024; Martin et al, 2018). The discovery of two pattern recognition receptors, cGLR1 and cGLR2, sensing dsRNA and triggering production of CDNs activating dSTING and Relish, provided strong support for the existence of two different pathways, activated by bacterial PGN

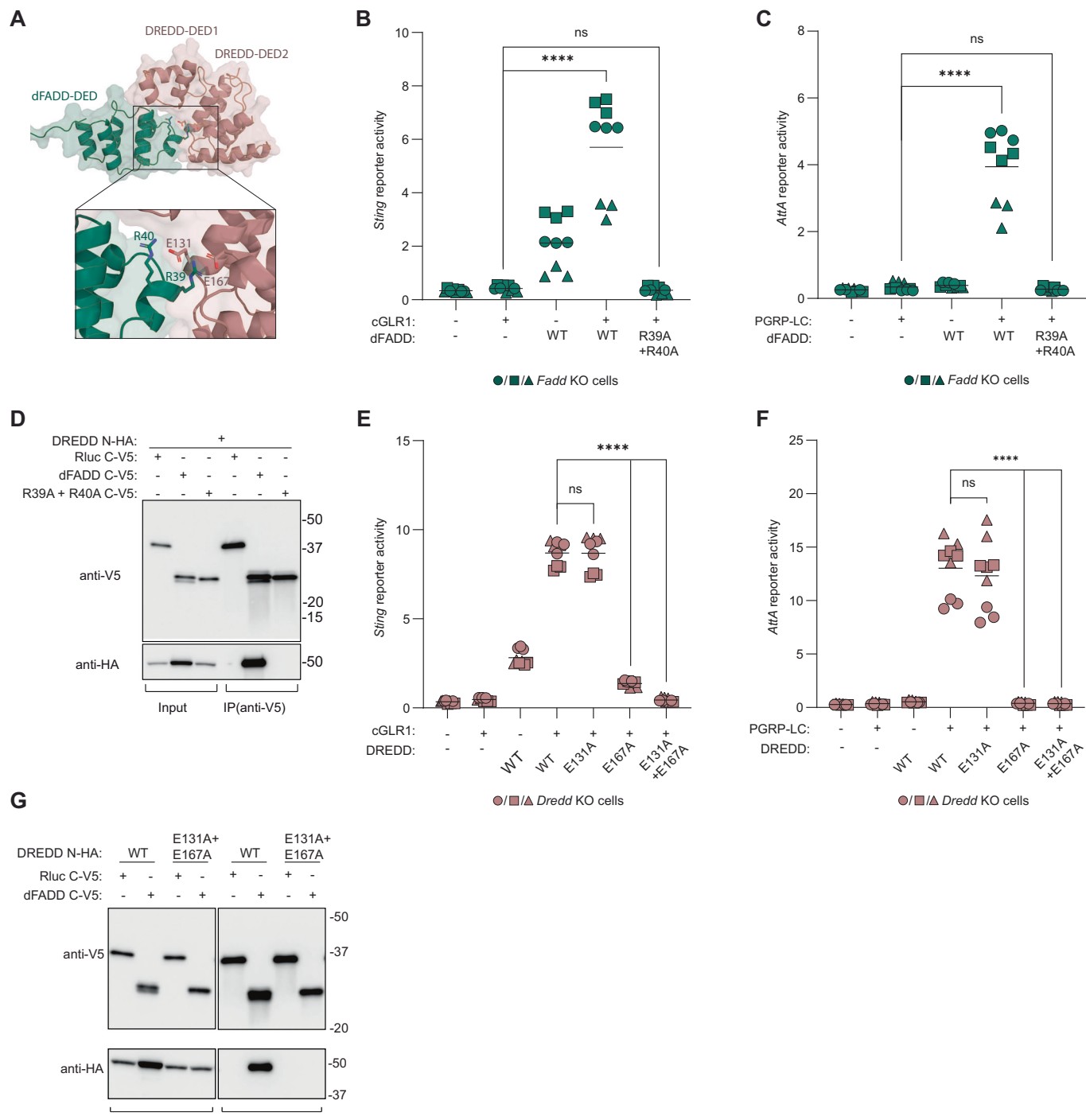

**Figure 4. dFADD recruits DREDD to the dSTING signalosome.**

(**A**) In silico prediction of the dFADD (green) and DREDD (pink) death effector domain (DED) complex. The insert shows key interacting residues. (**B, C**) Activation of *Sting* reporter upon cGLR1 expression (**B**) or *AttA* reporter upon PGRP-LC expression (**C**) in *Fadd* KO S2 cells reconstituted with WT dFADD or a mutant disrupting the predicted dFADD:DREDD interface. Data from three independent experiments (different geometrical icons), each performed in biological triplicate, are shown with mean ($n = 9$). $p$ values were calculated using two-way ANOVA, corrected with Dunnett's post hoc test: ****$p < 0.0001$, ns: $p = 0.6517$ (**B**) or 0.5948 (**C**), (**D**) Co-Immunoprecipitation of dFADD and DREDD or indicated mutants expressed in WT S2 cells. Representative of $n = 3$ independent experiments. (**E, F**) Activation of the *Sting* reporter upon cGLR1 expression (**E**) or *AttA* reporter upon PGRP-LC expression (**F**) in *Dredd* KO S2 cells reconstituted with WT DREDD or mutants disrupting the predicted dFADD:DREDD interface. Data from three independent experiments (different geometrical icons), each performed in biological triplicate, are shown with mean ($n = 9$). $p$ values were calculated using two-way ANOVA, corrected with Dunnett's post hoc test: ****$p < 0.0001$, ns: 0.9995 (**E**) or 0.1986 (**F**). (**G**) Co-immunoprecipitation of dFADD and DREDD or indicated mutants expressed in WT S2 cells. Source data are available online for this figure.

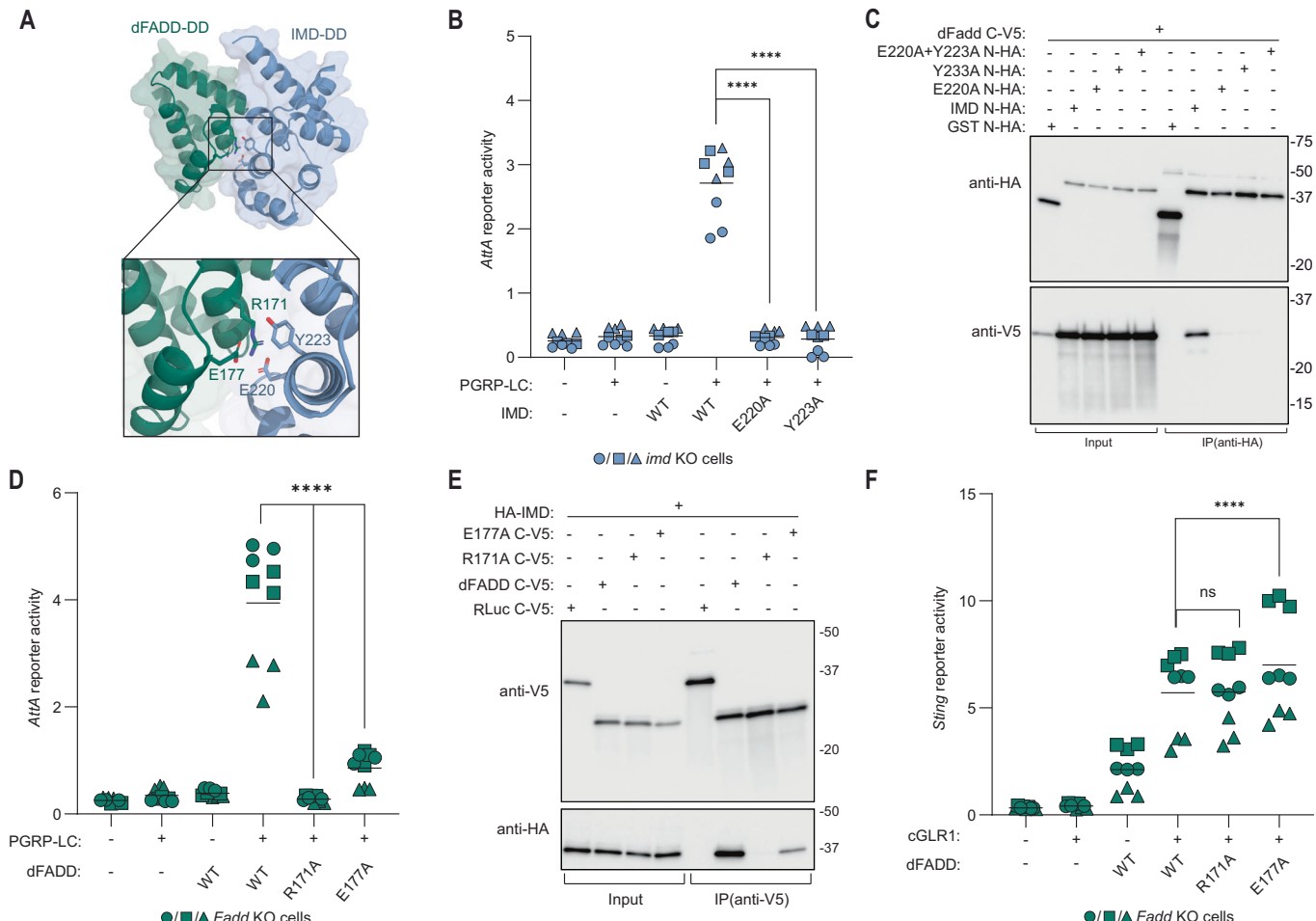

**Figure 5. dFADD interacts with IMD via its death domain, but these interactions are structurally distinct from the interactions with activated dSTING.**

(A) In silico prediction of the IMD (blue) and dFADD (green) death domain (DD) complex. The insert shows key interacting residues. (B) Induction of the *AttA* reporter in *imd* KO S2 cells upon expression of PGRP-LC and WT IMD or mutants disrupting the predicted IMD:dFADD interface. (C) Co-immunoprecipitation of WT IMD and WT dFADD or indicated IMD mutants ectopically expressed in WT S2 cells. Representative of $n = 3$ independent experiments. (D) Induction of the *AttA* reporter in *Fadd* KO S2 cells upon PGRP-LC expression and reconstitution with WT dFADD or indicated mutants disrupting the predicted IMD:dFADD interface. (E) Co-immunoprecipitation of WT IMD and WT dFADD or indicated dFADD mutants ectopically expressed in WT S2 cells. Representative of $n = 3$ independent experiments. (F) Induction of the *Sting* reporter in *Fadd* KO S2 cells upon expression of cGLR1 and WT dFADD or mutants disrupting the predicted IMD:dFADD interface. In (B, D, and F), data from three independent experiments (different geometrical icons), each performed in biological triplicate, are shown with mean ($n = 9$). $p$ values were calculated using two-way ANOVA, corrected with Dunnett's post hoc test: ****$p < 0.0001$, ns: $p = 0.9938$. Data in (D, F) were derived from the same experiments that are shown in Fig. 4c, b, respectively. Source data are available online for this figure.

and viral RNAs, respectively (Holleufer et al, 2021; Slavik et al, 2021). Yet, the question of how STING activates NF-κB in *Drosophila* and if this mechanism involves other components shared with the IMD pathway remained open. In particular, we initially found no effect on dSTING signaling in S2 cells when expression of *Fadd* or *Dredd* was silenced by RNA interference, although IMD signaling was impaired(Goto et al, 2018). We note, however, that at the same time others reported involvement of both DREDD and Relish in STING signaling in the silkworm *Bombyx mori* (Hua et al, 2018). Here, we took advantage of CRISPR engineering to create KO cell lines for these genes. Our data, which include complementation and was confirmed in vivo using mutant fly lines, clearly demonstrate the requirement for dFADD and DREDD in the dSTING pathway. Furthermore, we used a cellular model devoid of the confounding effects of microbiota on the IMD

pathway to clearly show that IMD and dSTING independently recruit dFADD and DREDD and that this leads to cleavage of Relish downstream of these two key signaling proteins. We hypothesize that our initial failure to detect the involvement of dFADD and DREDD in dSTING signaling using RNA silencing may be due to a long half-life of the proteins. Altogether, our data show that (i) dFADD and DREDD, in addition to the kinase dIKKβ and Relish, are shared between the IMD and dSTING pathways, and (ii) that dSTING and IMD can recruit dFADD independently of each other. Once recruited to either the IMD or dSTING signalosome, Relish is cleaved by the Caspase 8 homolog DREDD.

A remaining key question is how the IMD and dSTING pathways regulate different functional responses with the same transcription factor, Relish? A first hypothesis is that Relish may be differentially phosphorylated in the two pathways. Indeed, our

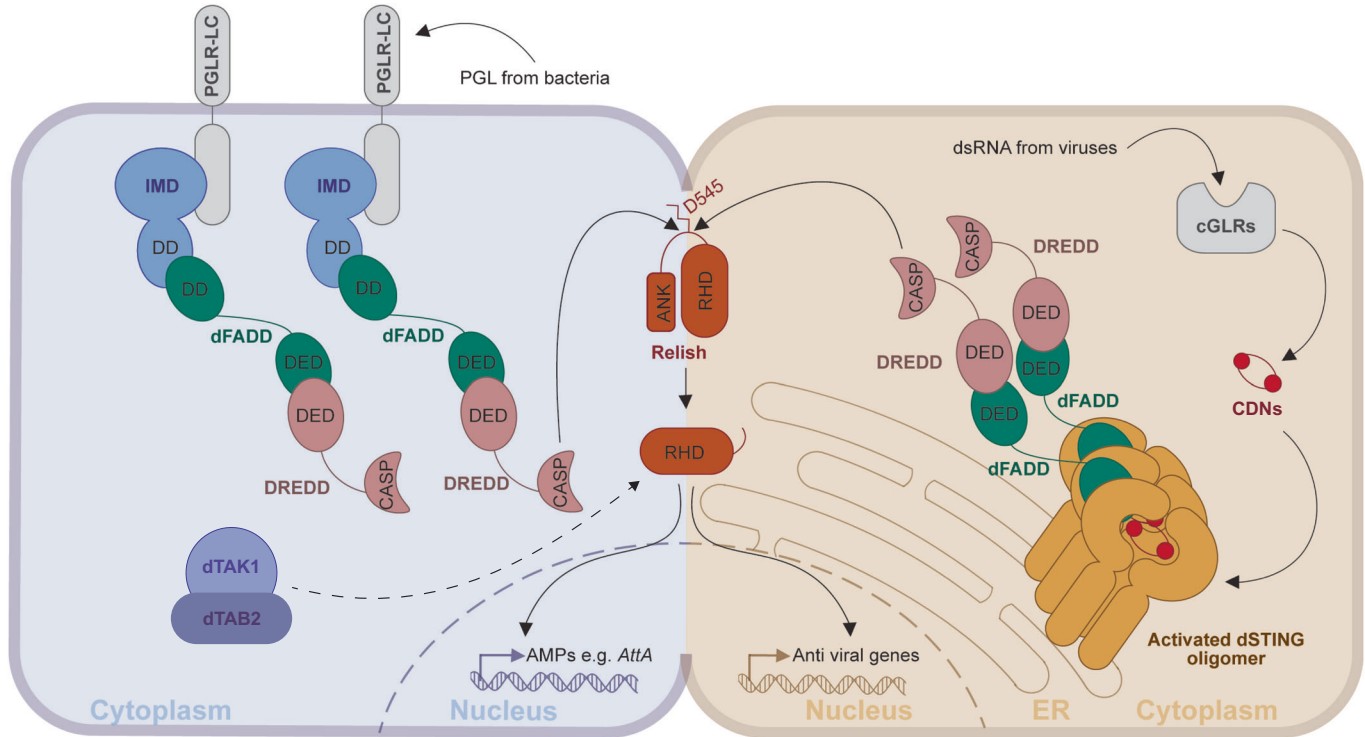

**Figure 6.  Overview of the IMD and cGLR-dSTING pathways.**

IMD and dSTING are independent activators of the NF-κB-like transcription factor Relish. The IMD pathway is activated when peptidoglycan (PGL) from bacteria are recognized by transmembrane PGRP-LC, while the dSTING pathway is activated when double-stranded RNA (dsRNA) from viruses are recognized by the cytoplasmic cGLR sleading to production of cyclic dinucleotides (CDNs). Both IMD and dSTING recruit the adapter molecule dFADD in complex with the Caspase-8 homologue DREDD, an interaction mediated by the death eff ector domain (DED) of both proteins. This allows the caspase domain (CASP) of DREDD to cleave Relish at the residue D545 thereby removing the inhibitory ankyrin repeat domain (ANK). Subsequently, the Rel homology domain (RHD) of Relish translocates to the nucleus to induce immune signaling. The interaction between dFADD and IMD is mediated by the death domain (DD) in both proteins, while the interaction between dFADD and dSTING are mediated by non-canonical interactions between the DD of dFADD and the oligomerized form of dSTING. Moveover, the dTAK1/dTAB2 complex is essential for IMD but not dSTING signaling.

observation that the kinase dTAK1, which is required for efficient IMD induction of antimicrobial peptides (Paquette et al, 2010), is not required for dSTING signaling suggests that dTAK1 recruitment might be an IMD-specific event, contributing to an IMD-specific signal output. Alternatively, a second hypothesis is that Relish partners with different transcription factors depending on which pathway is activated. The fact that members of the NF-κB family of transcription factors normally operate as either hetero- or homodimers (Tanji et al, 2010; Morris et al, 2016; Zhang et al, 2017), supports a role for a potential dimerization partner of Relish in functionally differentiating between the IMD and dSTING signaling output. Finally, a certain degree of crosstalk between the two pathways is likely to exist in vivo.

## A CTT-independent mechanism for dFADD recruitment to dSTING

dSTING oligomerization is a key event of its activation, but how are the various components of the NF-κB signaling machinery specifically recruited to oligomeric dSTING? We suggest an unexpected mechanism whereby this happens. Our data suggests a direct interaction between dSTING and dFADD. This is surprising as dSTING contains no death domain and hence cannot

form prototypic death domain interactions with dFADD. Yet, using AlphaFold Multimer, we produced a computational model showing how dFADD interacts specifically with the activated oligomeric form of dSTING, thus providing a clear mechanistic explanation for how dFADD is recruited specifically to the activated form of dSTING (Fig. 6).

We observe that it is necessary to mutate both dSTING interaction sites in dFADD to fully abrogate signaling. If one functional binding site remains in dFADD, this can bind to one dSTING dimer and will position the second binding site ideally for interacting with the second dimer of dSTING. The single amino acid mutation that we introduce into dFADD lowers binding affinity but does not completely abolish binding. We believe that the positional effect, possibly combined with the use of an overexpression system, masks the effect of a single mutation. The interaction between dFADD and dSTING is structurally distinct from the prototypic death domain interactions, which connect dFADD to IMD, but functionally similar since dFADD recruits DREDD and positions it for cleaving Relish in both signaling pathways.

In vertebrates, STING contains a C-terminal tail (CTT) composed of an unstructured stretch of 40–50 amino acids, the emergence of which coincided with the development of IFN

signaling (Margolis et al, 2017). This CTT is composed of discrete motifs mediating recruitment of kinases and transcription factors for activation, such as the pLxIS motif, mediating recruitment of IRF3 upon phosphorylation of the serine residue (Liu et al, 2015). In addition, the STING proteins from zebrafish and other ray-finned fish species contain a highly conserved PxExxD motif absent in the CTT of mammalian STING, which enables them to recruit TNF Receptor-Associated Factor 6 (TRAF6), resulting in enhanced NF-κB signal activation (de Oliveira Mann et al, 2019). Along the same line, bat species have lost the highly conserved serine residue at position 358 in human STING, dampening their capacity to activate synthesis of IFN (Xie et al, 2018). These data revealed that the CTT acts as a signaling platform whose modular architecture can gain or lose discrete motifs over evolutionary time to rewire immune responses. The dFADD interaction site we identified in dSTING is structurally and functionally distinct from the CTT found in vertebrate STING molecules and thus represents a different example of a mechanism for STING-dependent NF-κB activation. Further studies will be required to test if the recruitment of FADD to STING oligomers represents an ancestral mechanism of NF-κB activation, used by other invertebrates, e.g., sea anemones.

### The FADD-caspase 8 module in antiviral signaling

The versatile FADD-Caspase 8 axis is widely used in different immune signaling pathways throughout the animal kingdom. In mammals, this module has been implicated in inflammatory signaling and regulation of programmed cell death, through its recruitment by receptors of the tumor necrosis family and receptor-interacting serine/threonine-protein kinase (RIPK)1 and RIPK3 (Mouasni and Tourneur, 2018). Interestingly, a role for mammalian FADD in STING signaling was suggested in the first report identifying STING as a key regulator of innate immune signaling in mammals (Ishikawa and Barber, 2008), although little follow-up is available in the literature. Intriguingly, a role for FADD/Caspase 8 in antiviral immunity and IFN induction was proposed as early as 2004 (Balachandran et al, 2004). Yet, we find that the amino acids in dSTING, which are involved in the dFADD interaction, are only conserved among invertebrate STING and not vertebrate STING. Subsequent studies confirmed a role of FADD and Caspase 8 in inflammatory signaling, including response to the synthetic dsRNA analogue poly(I:C), although the underlying mechanisms are still poorly understood and the response to typical RIG-I activating viruses remained largely intact in *FADD* deficient cells (Balachandran et al, 2007; Kawai et al, 2005; Seth et al, 2005; Takahashi et al, 2006; Yoneyama et al, 2005). Altogether, these previous studies and our findings attest a central role of the FADD-Caspase 8 module in the control of transcriptional responses to viral infection in invertebrate and vertebrate animals.

In summary, our data show that activation of the dSTING and IMD pathways leads to the formation of independent and structurally distinct signalosomes. We explain the molecular basis for the specific recruitment of dFADD to activated oligomeric dSTING, bringing along the Caspase 8 homolog DREDD. Recruitment of the dFADD:DREDD complex to activated dSTING positions DREDD for cleavage of the NF-κB-like transcription factor Relish, constituting a key event in activation of Relish (Fig. 6).

## Methods

### Reagents and tools table

| Reagent/resource | Reference or source | Identifier or catalog number |
|---|---|---|
| **Experimental models** | | |
| Schneider 2 (S2) cells | Edzard Spillner | N/A |
| *D. melanogaster Relish* mutant flies | Professor Neal Silverman | *Relish^D545A* |
| *D. melanogaster Fadd* KO flies | Exelixis collection, Harvard Medical School | *Fadd^f02805* |
| *D. melanogaster w^1118* | IBMC | DrosDel *w^1118* |
| *D. melanogaster Dredd* KO flies | Professor Bruno Lemaitre | *Dredd^D55* |
| *D. melanogaster yw* | IBMC | N/A |
| *Tak* KO flies | This study | N/A |
| Knockout S2 cell lines | This study | N/A |
| **Recombinant DNA** | | |
| pAc/sgRNA-Cas9 | Addgene | 49330 |
| pGL3/*AttA* | Tauszig et al, 2000 | https://doi.org/10.1073/pnas.180130797 |
| pGLR3/*Sting* | Goto et al, 2018 | https://doi.org/10.1016/j.immuni.2018.07.013 |
| pRL | Goto et al, 2018 | https://doi.org/10.1016/j.immuni.2018.07.013 |
| pAc5.1/EGFP | Holleufer et al, 2021 | https://doi.org/10.1038/s41586-021-03800-z |
| pAc5.1/cGAS [155-522] | Holleufer et al, 2021 | https://doi.org/10.1038/s41586-021-03800-z |
| Additional plasmids and more information | This study | N/A |
| **Antibodies** | | |
| Mouse anti-V5-HRP | Invitrogen | 10402262 |
| Mouse anti-V5-HRP | Invitrogen | R96125 |
| Rabbit anti-myc | Abcam | ab9106 |
| Rat anti-HA-HRP | Roche | 12013819001 |
| mouse anti-actin | EMD Millipore | MAB1501 |
| rabbit anti-histone-H3 | Abcam | ab1791 |
| anti-mouse-HRP | Jackson ImmunoResearch | 715-036-150 |
| anti-rabbit-HRP | Cytiva | NA934 |
| **Oligonucleotides and other sequence-based reagents** | | |
| qPCR primers | Cai et al, 2020 | https://doi.org/10.1126/scisignal.abc4537 |

| Reagent/resource | Reference or source | Identifier or catalog number | Reagent/resource | Reference or source | Identifier or catalog number |
|---|---|---|---|---|---|
| *Fadd* sgRNA1 fw | This study | ttcgGATTGGTTCGC GACGCAGAT | *Sting* sgRNA5 rv | This study | aacTGTTCTTGGC TGATCTGCTCc |
| *Fadd* sgRNA1 rv | This study | aacATCTGCGTCGC GAACCAATCc | *Sting* sgRNA6 fw | This study | ttcgGTATCCAGTGCG AGTAGCTC |
| *Fadd* sgRNA2 fw | This study | ttcgGTTCTCGGTGC ATCCATCAA | *Sting* sgRNA6 rv | This study | aacGAGCTACTCGC ACTGGATACc |
| *Fadd* sgRNA2 rv | This study | aacTTGATGGATGC ACCGAGAACc | *ird5* sgRNA1 fw | This study | ttcgGACAATATA GTAATCCAACG |
| *Fadd* sgRNA3 fw | This study | ttcgGCAGTCAATT AAATCCTCGA | *ird5* sgRNA1 rv | This study | aacCGTTGGATTA CTATATTGTCc |
| *Fadd* sgRNA3 rv | This study | aacTCGAGGATTTA ATTGACTGCc | *ird5* sgRNA2 fw | This study | ttcgGTGCTGGAA TACTGTAACGG |
| *Rel* sgRNA1 fw | This study | ttcgGGTGGCACAG TGGCCGGAGC | *ird5* sgRNA2 rv | This study | aacCCGTTACAGT ATTCCAGCACc |
| *Rel* sgRNA1 rv | This study | aacGCTCCGGCCA CTGTGCCACCc | *ird5* sgRNA3 fw | This study | ttcgGTCTGACGG AGTTCGAGGTG |
| *Rel* sgRNA2 fw | This study | ttcgGAACAAACTG CACCGGATGA | *ird5* sgRNA3 rv | This study | aacCACCTCGAAC TCCGTCAGACc |
| *Rel* sgRNA2 rv | This study | aacTCATCCGGTG CAGTTTGTTCc | *key* sgRNA1 fw | This study | ttcgGGAACTGTCC GGCATGAGCG |
| *Rel* sgRNA3 fw | This study | ttcgGCTGCGGATC GTTGAGCAAC | *key* sgRNA1 rv | This study | aacCGCTCATGCC GGACAGTTCCc |
| *Rel* sgRNA3 rv | This study | aacGTTGCTCAAC GATCCGCAGCc | *key* sgRNA2 fw | This study | ttcgGAAGAGTCA TTCGTTATCTT |
| *Dredd* sgRNA4 fw | This study | ttcgGATCAGATC GTTCTGATCGA | *key* sgRNA2 rv | This study | aacAAGATAACGA ATGACTCTTCc |
| *Dredd* sgRNA4 rv | This study | aacTCGATCAGAA CGATCTGATCc | *key* sgRNA3 fw | This study | ttcgGGAATACAGT TCACTGTTGG |
| *Dredd* sgRNA5 fw | This study | ttcgGCTTTATGGC GACGACCACT | *key* sgRNA3 tv | This study | aacCCAACAGTGA ACTGTATTCCc |
| *Dredd* sgRNA5 rv | This study | aacAGTGGTCGTC GCCATAAAGCc | *key* sgRNA4 fw | This study | ttcgGACGCTCAC CTTCAGGACAT |
| *Dredd*sgRNA6 fw | This study | ttcgGGCCTCTGC TTTCTGCTTTA | *key* sgRNA4 rv | This study | aacATGTCCTGAA GGTGAGCGTCc |
| *Dredd* sgRNA6 rv | This study | aacTAAAGCAGA AAGCAGAGGCCc | *key* sgRNA5 fw | This study | ttcgGTACTGGTG TTCTCGGTCGG |
| *Dredd* sgRNA7 fw | This study | ttcgGTGGTCTTC ATCCTGAGCCA | *key* sgRNA5 rv | This study | aacCCGACCGAGA ACACCAGTACc |
| *Dredd* sgRNA7 rv | This study | aacTGGCTCAGG ATGAAGACCACc | *imd* sgRNA2 fw | This study | ttcgCATCGAGCA GGCGCACATCC |
| *Dredd* sgRNA8 fw | This study | ttcgGTGCTGGTC CGGCGACACGG | *imd* sgRNA2 rv | This study | aacGGATGTGCG CCTGCTCGATGc |
| *Dredd* sgRNA8 rv | This study | aacCCGTGTCGCC GGACCAGCACc | *imd* sgRNA3 fw | This study | ttcgCAGCAGTG TAGTAAGTCGTC |
| *Dredd* sgRNA9 fw | This study | ttcgGATGCTGTT GGATGCGTAGA | *imd* sgRNA3 rv | This study | aacGACGACTTA CTACACTGCTGc |
| *Dredd* sgRNA9 rv | This study | aacTCTACGCAT CCAACAGCATCc | *imd* sgRNA4 fw | This study | ttcgGGTCAGATC CGAGGAGGCTG |
| *Sting* sgRNA4 fw | This study | ttcgGGTGGCTAC AATGCGAATAG | *imd* sgRNA4 rv | This study | aacCAGCCTCCTC GGATCTGACCc |
| *Sting* sgRNA4 rv | This study | aacCTATTCGCAT TGTAGCCACCc | *imd* sgRNA5 fw | This study | ttcgGTCGCTGAG CTCGCGCAGCA |
| *Sting* sgRNA5 fw | This study | ttcgGAGCAGATC AGCCAAGAACA | *imd* sgRNA5 rv | This study | aacTGCTGCGCG AGCTCAGCGACc |

| Reagent/resource | Reference or source | Identifier or catalog number |
|---|---|---|
| *imd* sgRNA6 fw | This study | ttcgGAACCAGATAACAACAACAG |
| *imd* sgRNA6 rv | This study | aacCTGTTGTTGTTATCTGGTTCc |
| *imd* seq fw | This study | CGAGCAGCATGTCAAAGCTC |
| *imd* seq rv | This study | CTCCAGTGCCTTCCAAACCA |
| *Sting* seq fw | This study | TTCGCGAGATCTCCAAATCG |
| *Sting* seq rv | This study | AGGGGCTAGAAAACAAAATGCT |
| *Dredd* seq fw | This study | GGAAAGTACACGTGCTGGCG |
| *Dredd* seq rv | This study | GCCACGGCTATCGGATGTCA |
| *Rel* seq fw | This study | ACACTCTTTCCCTACACGACGCTCTTCCGATCTGCGTTAGTTTCGGCGTTGCT |
| *Rel* seq rv | This study | GACTGGAGTTCAGACGTGTGCTCTTCCGATCTGACACGTGCAAATCATGCGGA |
| *Fadd* seq fw | This study | ACACTCTTTCCCTACACGACGCTCTTCCGATCTCAGGCACTGGAGCTACGACA |
| *Fadd* seq rv | This study | GACTGGAGTTCAGACGTGTGCTCTTCCGATCTCTACAGCGGCAGCTAATTCCGA |
| **Chemicals, Enzymes and other reagents** | | |
| Penicillin-streptomycin | Gibco™ | Cat#15140-122 |
| Fetal bovine serum | Bio&Sell | FBS.S.0615 HI |
| Schneider's Drosophila Medium | Biowest | L0207-500 |
| Passive Lysis Buffer | Promega | E194A |
| Effectene Transfection Reagent | Qiagen | 301425 |
| Dual-Luciferase® Reporter Assay System | Promega | E1980 |
| cOmplete™, EDTA-free Protease Inhibitor Cocktail | Roche | 5056489001 |
| V5-Trap Agarose beads | ChromoTek | v5ta-20 |
| anti-HA-Agarose beads | Sigma-Aldrich | A2095-1ML |
| Trans-Blot nitrocellulose membranes | Bio-Rad | 5671104 |
| iBlot™ 3 Mini PVDF Transfer Stacks | Thermo Fisher Scientific | IB34002 |

| Reagent/resource | Reference or source | Identifier or catalog number |
|---|---|---|
| n-dodecyl-β-D-maltoside (DDM) | Inalco | 1758-1350 |
| Kaleidoscope™ | Bio-Rad | 1610375 |
| Anti-V5 Agarose Affinity Gel antibody | Sigma-Aldrich | A7345 |
| 1.9 µm C18 beads | Dr. Maisch, GmbH | |
| JetOPTIMUS® | Polyplus | 117-15 |
| QIAamp DNA Blood Mini Kit | Qiagen | 51104 |
| E.Z.N.A. Tissue DNA Kit | Omega Bio-tek | D3396 |
| CloneJET PCR Cloning Kit | Thermo Fisher Scientific | K1231 |
| 2′3′-cGAMP | Biolog | C161 |
| 3′2′-cGAMP | Biolog | C328 |
| *Serratia marcescens* (*S.m*) culture | Deng et al, 2022 | https://doi.org/10.3389/fimmu.2022.933137 |
| TRIzol | Invitrogen | 15596078 |
| iScript gDNA Clear cDNA Synthesis Kit | Bio-Rad | 1725035 |
| SYBR Green master mix | Accurate Biotechnology (Hunan) | Code. AG11701 |
| Drosophila C virus (DCV) | Sabatier et al, 2003 | https://doi.org/10.1046/j.1432-1033.2003.03725.x |
| **Software** | | |
| GraphPad software | https://www.graphpad.com/ | Version 10.2.3 |
| R | https://www.r-project.org/ | Version 4.3.3 |
| PyMOL | https://www.pymol.org/ | Version 3.0.1 |
| Coot | https://www2.mrc-lmb.cam.ac.uk/personal/pemsley/coot/ | Version 0.8.9 |
| Adobe Illustrator | https://www.adobe.com/products/illustrator.html | Version 28.5.0 |
| Proteome Discoverer | https://www.thermofisher.com/dk/en/home/industrial/mass-spectrometry/liquid-chromatography-mass-spectrometry-lc-ms/lc-ms-software/multi-omics-data-analysis/proteome-discoverer-software.html | Version 2.5 |
| PAE Viewer | Elfmann and Stülke, 2023 https://pae-viewer.uni-goettingen.de/ | https://doi.org/10.1093/nar/gkad350 |
| **Other** | | |
| Varioskan LUX microplate reader | Thermo Fisher Scientific | N/A |
| 96-wells plates | Thermo Fisher Scientific | N/A |
| GloMax® microplate reader | Promega | N/A |
| Bioruptor Plus | Diagenode | N/A |

| Reagent/resource | Reference or source | Identifier or catalog number |
|---|---|---|
| Fisherbrand™ Elmasonic Select 30 | Thermo Fisher Scientific | N/A |
| Orbitrap Eclipse Tribrid mass spectrometer | Thermo Fisher Scientific | N/A |
| EASY 1200 nLC | Thermo Fisher Scientific | N/A |
| Sequencing | Eurofins Genomics | N/A |
| Protein sequence (dSTING) | uniport.org | UniprotID: A0A0B4LFY9 |
| Protein sequence (dFADD) | uniport.org | UniprotID: Q9V3B4 |
| Protein sequence (DREDD) | uniport.org | UniprotID: Q8IRY7 |
| Protein sequence (IMD) | uniport.org | UniprotID: Q7K4Z4 |
| UniProt *D. melanogaster* reference proteome | uniport.org | v2023-06-28 |

## Methods and Protocols

### Plasmids

Suitable target sequences for KO were predicted using E-CRISP (Heigwer et al, 2014) and cloned into pAc/sgRNA-Cas9 (Addgene: 49330) as annealed oligoes, like in (Bassett et al, 2014). *Sting*, *Fadd*, *Dredd*, *imd*, *PGRP-LC*, *cGlr1*, and *Rel* were cloned from S2 cells and ligated into the pAc5.1 expression vector encoding either an N-terminal HA-tag, C-terminal V5-tag or C-terminal myc-tag. Indicated mutations were introduced by site-directed mutagenesis. Please note that the *Fadd* sequence cloned from S2 cells contains the following non-silent polymorphisms: I71M, K87P, T135P, P143T and R232K. pGL3/*AttA* encoding the firefly luciferase gene under transcriptional control of the 2.3 kb proximal region of the *AttA* promoter, and the pGL3/*Sting* reporter encoding the firefly luciferase gene under transcriptional control of the 200 bp proximal region of the *Sting* promoter, and pRL constitutively expressing *Renilla* luciferase were previously described (Goto et al, 2018).

## Culture of S2 cells

S2 cells were gifted by Edzard Spillner. Cells were cultured in Schneider's Drosophila Medium supplemented with 1% penicillin-streptomycin and 10% fetal bovine serum (Gibco™) at 26 °C.

## Dual-luciferase assay

Cells used for IMD or dSTING dual-luciferase reporter assays were seeded at $1.5 \times 10^6$ cells/mL in 24-well plates and transfected with either 50 ng pGL3/*AttA* or 242.5 ng pGL3/*Sting* reporter, 15 ng pRL-actin, 50 ng pAc5.1/cGLR1-C-myc or pAc5.1/PGRP-LCa-C-myc, and empty pAc5.1 plasmid (Bonnay et al, 2014) up to a total amount of 0.5 μg using JetOPTIMUS® DNA transfection reagent (PolyPlus) following the manufacturer's instructions. Additionally, 0.5 ng pAc5.1/N-HA-IMD, 10 ng pAc5.1/dSTING-C-V5, 0.5 ng

pAc5.1/N-HA-DREDD or 1 ng pAc5.1/N-HA-dFADD were co-transfected as indicated. For immunoblotting of IMD, DREDD, dSTING, or dFADD, cells were transfected with increased amounts of pAc5.1/N-HA-IMD (200 ng), pAc5.1/N-HA-DREDD (200 ng), pAc5.1/dSTING-C-V5 (750 ng), and pAc5.1/N-HA-dFADD (200 ng) in additional experimental conditions.

Twenty-four hours post-transfection, cells were collected and lysed in 100 μL passive lysis buffer (Promega) per well and centrifuged at 18,000×*g* for 5 min. About 10 μL cleared lysate were transferred to white 96-well plates (Thermo Fisher Scientific) for measurement of firefly and *Renilla* luciferase activity with Dual-Luciferase® Reporter Assay System (Promega) according to the manufacturer's instructions. Measurements were made with either a Varioskan LUX microplate reader (Thermo Fisher Scientific) with a 1 s acquisition time or a GloMax® microplate reader (Promega) with a 0.3 s acquisition time. Prior to immunoblotting, cells were lysed directly in 28 μL protein sample buffer (100 mM Tris-HCl, [pH 6.8], 4% SDS, 20% glycerol, 0.2 M DTT, 0.5% bromophenol blue) per well.

## Relish cleavage

To assess proteolytic cleavage of Relish, S2 cells were transfected with 100 ng pAc5.1/Relish C-V5 as described above. Three wells were transfected for each experimental condition and lysed after 24 h in 74 μL protein sample buffer (100 mM Tris-HCl, [pH 6.8], 4% SDS, 20% glycerol, 0.2 M DTT, and 0.5% bromophenol blue) and subjected to immunoblotting.

## Immunoprecipitation

For co-immunoprecipitation, S2 cells were seeded in six-well plates ($3 \times 10^6$ cells/well, 2 wells/condition) and transfected with HA-tagged and V5-tagged proteins as indicated (50 ng–1 μg/well/plasmid) using the Effectene transfection reagent (Qiagen) according to the manufacturer's instructions. Cells were harvested two or three days post-transfection, washed once with ice-cold phosphate-buffered saline (1× PBS), and resuspended in 0.5 mL of NET buffer (50 mM Tris-HCl [pH 7.4], 150 mM NaCl, 1 mM EDTA, and 0.1% NP40), containing 5% glycerol and supplemented with 1x cOmplete™, EDTA-free Protease Inhibitor Cocktail (5056489001, Roche). Cells were lysed by three rounds of sonication for 30 s (Bioruptor Plus, Diagenode) and spun for 15 min at 16,000×*g* at 4 °C. An aliquot of the cleared lysates (50 μL) was set aside as protein input. The supernatant was then complemented with 500 μL of NET buffer and with either 20 μL of V5-Trap Agarose beads (v5ta-20, ChromoTek) or 40 μL of anti-HA-Agarose beads (A2095-1ML, Sigma-Aldrich), and samples were mixed for 2–4 h (12 rpm) on a rotating wheel at 4 °C. Sample beads were washed three times with 0.5 mL of NET buffer, followed by a fourth wash in NET buffer without NP40. Bound proteins were finally eluted with 80–100 μL of protein sample buffer (100 mM Tris-HCl, [pH 6.8], 4% SDS, 20% glycerol, 0.2 M DTT, 0.5% bromophenol blue) and subjected to immunoblotting.

## Immunoblotting

Lysates were incubated at 80 °C in a shaking incubator for 10 min and subsequently at 95 °C for 5 min. The samples were subjected to SDS-PAGE (Bio-Rad or Thermo Fisher Scientific) and transferred to Trans-

Blot nitrocellulose membranes (Bio-Rad) or iBlot™ 3 Mini PVDF Transfer Stacks (IB34002, Thermo Fisher Scientific) before blocking. Membranes were incubated with anti-V5-HRP (10402262, Invitrogen 1:5000 or R96125, Thermo Fisher Scientific), rabbit anti-myc (ab9106, Abcam, 1:2000), anti-HA-HRP (12013819001, Roche, 1:5000), mouse anti-actin (MAB1501, EMD Millipore, 1:1000), rabbit anti-histone-H3 (ab1791, Abcam, 1:5000), anti-mouse-HRP (715-036-150, Jackson ImmunoResearch, 1:15,000)), anti-rabbit-HRP (NA934, Cytiva, 1:10,000) and detected with a ChemiDoc Imaging System (Bio-Rad). Expression of cGLR1-C-myc or PGRP-LC-C-myc was verified on separate blots as indicated.

## Structure prediction using AlphaFold

A local installation of AlphaFold v2.1.0 (Jumper et al, 2021) were used for in silico prediction of dFADD in complex with dSTING, DREDD, or IMD. The following protein sequences used as input for AlphaFold were derived from UniProt: dSTING (A0A0B4LFY9), dFADD (Q9V3B4), DREDD (Q8IRY7), IMD (Q7K4Z4) (https://www.uniprot.org/). Primary accession numbers are shown in parentheses. All main figures of protein structures were created with PyMOL v. 2.5 (The PyMOL Molecular Graphics System, Version 2.5, Schrödinger, LLC). Extended data figures of protein structures and PAE plots were created with the online tool "PAE Viewer" (Elfmann and Stülke, 2023).

## dSTING immunoprecipitation and mass spectrometry

For immunoprecipitation/mass spectrometry, S2 cells were seeded in 6-well plates ($3 \times 10^6$ cells/well, 4 wells/conditions, 2.5 mL/well). After 24 h, the cells were transfected with pAc5.1/dSTING-C-V5 (17.5 μg/well), and pAc5.1/cGAS[155–522] (7.5 μg/well) or empty pAc5.1 (7.5 μg/well) as described above. A control condition with DNA encoding EGFP (25.0 μg/well) was included. Cells were harvested 24 h post-transfection, washed once with ice-cold phosphate-buffered saline (PBS), and resuspended in 1 mL NET buffer (50 mM Tris-HCl, [pH 7.4], 150 mM sodium chloride (NaCl), 1 mM Ethylenediaminetetraacetic acid (EDTA), 1% dodecyl-β-D-maltoside (DDM), 10 μM iodoacetamide (IAA), 10% glycerol) and supplemented with 1x cOmplete™, EDTA-free Protease Inhibitor Cocktail (5056489001, Roche). Cells were lysed by 30 min rotation (2 rpm) at 4 °C, followed by $3 \times 30$ s sonication (Fisherbrand™ Elmasonic Select 30, Thermo Fisher Scientific). The lysates were cleared by 15 min. centrifugation at 17,000×$g$, 4 °C. About 100 μL Anti-V5 Agarose Affinity Gel antibody (A7345, Sigma-Aldrich) were added to each sample, and the volume was adjusted to 2 mL with buffer. Samples were rotated for 4 h (2 rpm) at 4 °C to allow immunoprecipitation. The beads were washed four times with 1 mL of NET buffer (50 mM Tris-HCl [pH 7.4], 150 mM sodium chloride (NaCl), 1 mM ethylenediaminetetraacetic acid (EDTA), 1% dodecyl-β-D-maltoside (DDM), 10 μM iodoacetamide (IAA), 10% glycerol) and supplemented with 1x cOmplete™, EDTA-free Protease Inhibitor Cocktail (5056489001, Roche) before bound proteins were eluted with 200 μL 100 mM glycine buffer (pH 2.5) per sample at RT for 10 min under rotation.

Samples were prepared for mass spectrometry by the filter-aided sample preparation method as described elsewhere (Wisniewski et al, 2009). The samples were analyzed on an Orbitrap Eclipse Tribrid mass spectrometer (Thermo Fisher Scientific) coupled to an EASY 1200 nLC (Thermo Fisher Scientific). Peptides were loaded onto a trap column (2 cm × 75 μm i.d.) and separated on an analytical column (20 cm × 75 μm i.d.) packed with 1.9 μm C18 beads (Dr. Maisch, GmbH). The samples were eluted at a flow rate of 250 nL/min and with a gradient from 5 to 35% acetonitrile for either 50 min or 80 min. followed by a steep increase to 80% acetonitrile for 10 min. Raw files were processed in Proteome Discoverer 2.5 using the Sequest HT search algorithm. The UniProt *D. melanogaster* reference proteome (v2023-06-28) was used as a database and the following search parameters: 10 ppm precursor mass accuracy, 0.02 Da fragment mass accuracy, trypsin as digestion enzyme, max. two missed cleavage sites, Oxidation (M) as dynamic modification and acetyl (N-term), Met-loss (M), and Met-loss+Acetyl (M) as dynamic protein N-terminus modifications. Carbamidomethyl (C) as static modification. Protein abundances were label-free quantified based on precursor area and on unique peptides. Data were filtered to only include proteins identified with a 1% FDR.

## Generation of S2 KO cell pools

S2 cells were transfected using JetOPTIMUS as described above with several pAc/sgRNA-Cas9 plasmids per gene, each encoding a sgRNA specific for the sequences (see Reagents and Tools Table). Seventy-two hours post-transfection, cells were selected with media supplemented with a range of puromycin from 0.5 to 15 μg/mL for 3 weeks. Cells surviving treatment with the highest concentration of puromycin were used for downstream applications.

To verify editing by Cas9, genomic DNA of KO cell pools and WT control cells were purified using QIAamp DNA Blood Mini Kit (QIAGEN) or E.Z.N.A Tissue DNA kit (D3396) following the manufacturer's instructions. Targeted regions were amplified utilizing gene-specific primers (see Reagents and Tools Table) before analysis on 1% agarose gels (all cell lines except *Tak1* and *Tab2* KO). The PCR products from the *Tab2* KO pool were verified directly by Sanger sequencing using the PCR primers (Eurofins Genomics), while the PCR products from the *Tak1* KO cell pool were cloned using CloneJET PCR Cloning Kit (ThermoScientific, #K1231) before verification with Sanger sequencing using the PCR primers (Eurofins Genomics).

## Maintaining flies and strains

Flies were maintained on standard cornmeal agar medium at 25 °C. *Relish^D545A* flies gifted by Professor Neal Silverman and *Fadd^f02805* (Exelixis collection, Harvard Medical School) were isogenized to the *w^1118* background. *Dredd^D55* were gifted by Professor Bruno Lemaitre and were isogenized to the yw background. All the flies used were *Wolbachia*-free.

## Gene induction in flies

For *Srg* induction, WT *w^1118* or *Relish^D545A* flies were intrathoracically injected with 69 nL of 0.9 ng/nL 2'3'-cGAMP (Biolog, C161) or 10 mM Tris, pH 7.5, as a negative control. *yw*, *yw^Dredd−/−*, *w^1118*, *w^1118 Fadd−/−*, or *w1118 Tak1−/−* flies were instead injected with 69 nL of 0.9 ng/nL 3'2'-cGAMP (Biolog, C328) or 10 mM Tris, pH 7.5 as negative control. For *AMP* induction, flies were injected

with 4.6 nL of a *Serratia marcescens* (*S.m*) culture with an optical density at 600 nm (OD600) of 1 (Deng et al, 2022). It was previously verified that the overall response of 2'3'-cGAMP and 3'2'-cGAMP is highly similar (Cai et al, 2020; Cai et al, 2023).

Six flies (three males and three females) were pooled and homogenized for RNA extraction using TRIzol (Invitrogen, 15596078) following the manufacturer's instructions. RNA was quantified by nanodrop, and 500 ng of RNA was used for reverse transcription with iScript gDNAclear cDNA synthesis kit (Bio-Rad) following the manufacturer's instructions.

RT-qPCR was performed using SYBR Green master mix (Bio-Rad) and 0.5 mM forward and reverse primers (see oligo list). The RT-qPCR program was the following: initial denaturation for 15 s at 98 °C followed by 35 cycles of 2 s at 95 °C and 30 s at 60 °C. The threshold cycle (Ct) of each sample was calculated by linear regression. Analysis was made by the ΔCt method using RpL32 as a reference gene: $_2Ct(RpL32)-Ct(target)$.

## Survival of flies

WT $w^{1118}$ or $Relish^{D545A}$ flies were intrathoracically injected with a total volume of 69 nL containing either 10 mM Tris, pH 7.5, as a negative control or 50 pfu DCV or 50 pfu DCV with 58 ng 2'3'-cGAMP. Survival of thirty flies (15 females and 15 males) was monitored daily for 20 days; a total of six such groups of flies were included in the presented data. Flies that did not survive the initial injection were removed from analysis.

WT *yw* and $yw^{Dredd-/-}$ flies or WT $w^{1118}$ and $w^{1118, Fadd-/-}$ flies were likewise injected with a total volume of 69 nL containing 50 pfu of DCV in 10 mM Tris, pH 7.5, with and without supplementation of 58 ng 3'2'-cGAMP. Survival of 60 flies (30 female, 30 male) were monitored daily for 20 days; a total of six such groups of flies were included in the presented data. Flies that did not survive initial injection were removed from analysis.

## Figures and software

Data were statistically analyzed in Prism version 10.2.3 (GraphPad software) or R version 4.3.3 and visualized in Prism. Protein structures were visualized in PyMOL version 3.0.1 and Coot 0.8.9. Illustrative figures were prepared in Illustrator 28.5.0 (Adobe). Mass spectrometry raw data were analyzed in Proteome Discoverer 2.5.

## Statistics

Dual luciferase assay experiments were independently repeated the stated number of times, with each experimental condition containing three technical replicates. Data were analyzed by a two-tailed ANOVA investigating the effect of experimental conditions across all experimental repeats. Experimental conditions to be included for multiple comparison were manually selected after data observation, and the resulting *p* value adjusted using the two-tailed Holm–Sidak correction. Normality was manually evaluated by QC plot analysis, and data were log transformed when this improved normality. Gene induction in flies by permutation analysis in R using the Coin package with Benjamini–Hochberg correction. Survival experiments were analyzed by a Gehan–Breslow–Wilcoxon test in Prism.

## Data availability

The mass spectrometry proteomics data have been deposited to the ProteomeXchange Consortium (https://www.proteomexchange.org/) via the PRIDE (Perez-Riverol et al, 2022) partner repository with the dataset identifier PXD054811. The generated AlphaFold model has been submitted to https://modelarchive.org/ with the accession codes ma-aqyll, (https://modelarchive.org/doi/10.5452/ma-aqyll), ma-hozlu (https://modelarchive.org/doi/10.5452/ma-hozlu), and ma-16ggp (https://modelarchive.org/doi/10.5452/ma-16ggp).

The source data of this paper are collected in the following database record: biostudies:S-SCDT-10_1038-S44318-026-00761-9.

## Peer review information

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

## Acknowledgements

We are grateful to Rune T. Kidmose, Esben Lorentzen, and Hans Henrik Gad for helpful discussions on the proper use of AlphaFold. RH was supported by The Novo Nordisk Foundation distinguished investigator grant, NNF23OC0082384; Independent Research Fund Denmark, Medical Science grant No. 2034-00225B and Natural Science, grant No. 0135-00338B. J-LI was supported by the following grants: ANR-22-CE15-0019, ANR-10-IDEX-0002, ANR-20-SFRI0012, ANR-17-EURE-0023, and acknowledged support from the Hoffmann Infinitus Program. J-LI and HC acknowledge support from the National Key R&D Program of China (2023YFE0107700). HC was supported by the National Science Foundation (3230767 and 32000662), Guangdong Provincial Science Fund for Distinguished Young Scholars (2023B1515020098), and Guangdong Provincial Young Scholars Academic Exchange Program (2022A0505030018). KGW and JS were both supported by PhD fellowships from MENRT and the Fondation pour la Recherche Médicale (KGW) and the Fondation ARC (JS). National Institute of Allergy and Infectious Diseases (NIAID), AI060025 (NS).

## Author contributions

**Kasper Grønbjerg Winther**: Conceptualization; Formal analysis; Investigation; Methodology; Writing—original draft. **Juliette Schneider**: Investigation; Methodology. **Gabrielle Haas**: Data curation; Investigation; Methodology; Project administration. **Anna Hvarregaard Christensen**: Data curation; Formal analysis; Validation; Investigation; Visualization; Methodology; Writing—original draft; Writing—review and editing. **Shreya Goyal**: Investigation. **Weisheng Luo**: Investigation. **Ziming Wei**: Investigation. **Jiyong Liu**: Investigation. **Katarzyna Kjøge**: Data curation; Investigation; Methodology. **Jan J Enghild**: Resources; Supervision. **Carine Meignin**: Supervision; Visualization. **Neal Silverman**: Resources; Supervision; Writing—review and editing. **Hua Cai**: Supervision; Funding acquisition; Investigation; Methodology; Project administration. **Jean-Luc Imler**: Conceptualization; Formal analysis; Supervision; Funding acquisition; Project administration; Writing—review and editing. **Rune Hartmann**: Conceptualization; Data curation; Formal analysis; Supervision; Funding acquisition; Writing—original draft; Project administration; Writing—review and editing.

Source data underlying figure panels in this paper may have individual authorship assigned. Where available, figure panel/source data authorship is listed in the following database record: biostudies:S-SCDT-10_1038-S44318-026-00761-9

## Disclosure and competing interests statement

The authors declare no competing interests.

# Expanded View Figures

**Figure EV1.  Signaling in S2 *Relish* KO pool and Srg-induction in *Relish^{D545A}* flies.**

(**A**) Induction of *AttA* luciferase reporter upon expression of PGRP-LC in WT (gray) or *Relish* KO S2 cells (brown). Data from three independent experiments (different geometrical icons), each performed in biological triplicate, are shown with mean ($n = 9$). *P* values were calculated using two-way ANOVA, corrected with Tukey's post hoc test: ****$p < 0.0001$, ns: $p > 0.9999$ (**B**) Western blots of cell lysates from (**A**). Dotted lines indicate separate gels. (**C**) Induction of *Sting* luciferase reporter upon expression of cGLR1 in WT (gray) or *Relish* KO S2 cells (brown). Data from three independent experiments (different geometrical icons), each performed in biological triplicates, are shown with mean ($n = 9$). *P* values were calculated using two-way ANOVA, corrected with Tukey's post hoc test: ****$p < 0.0001$, ns: $p > 0.9999$. (**D**) western blots of cell lysates from (**C**). Dotted lines indicate separate gels. (**E–G**) Induction of *Srg1*, *Srg2*, or *Srg3* in $w^{1118}$ (control) or *Relish^{D545A}* flies measured by qPCR 24 h after intrathoracic injection of the STING agonist 2'3'-cGAMP. Each data point is derived from a pool of six flies (three male, three female). Bars represent mean ± standard deviation. *P* values were calculated using a pairwise permutation test corrected with the Benjamini–Hochberg method: ***$p = 0.0008526$ (**F**) or 0.0006075 (**G**), **$p = 0.007433$, ns: $p = 0.8724$ (**E**), 0.7777 (**F**), or 0.1125 (**G**). (**H**) Survival of $w^{1118}$ (control) or *Relish^{D545A}* flies injected with Drosophila C virus (DCV) and co-injected with Tris or 2'3'-cGAMP. Points and bars represent mean ± standard error. *P* values were calculated with a Gehan–Breslow–Wilcoxon test: ****$p < 0.0001$. Main Fig. 1C shows a sub-portion of data from this figure. Source data are available online for this figure.

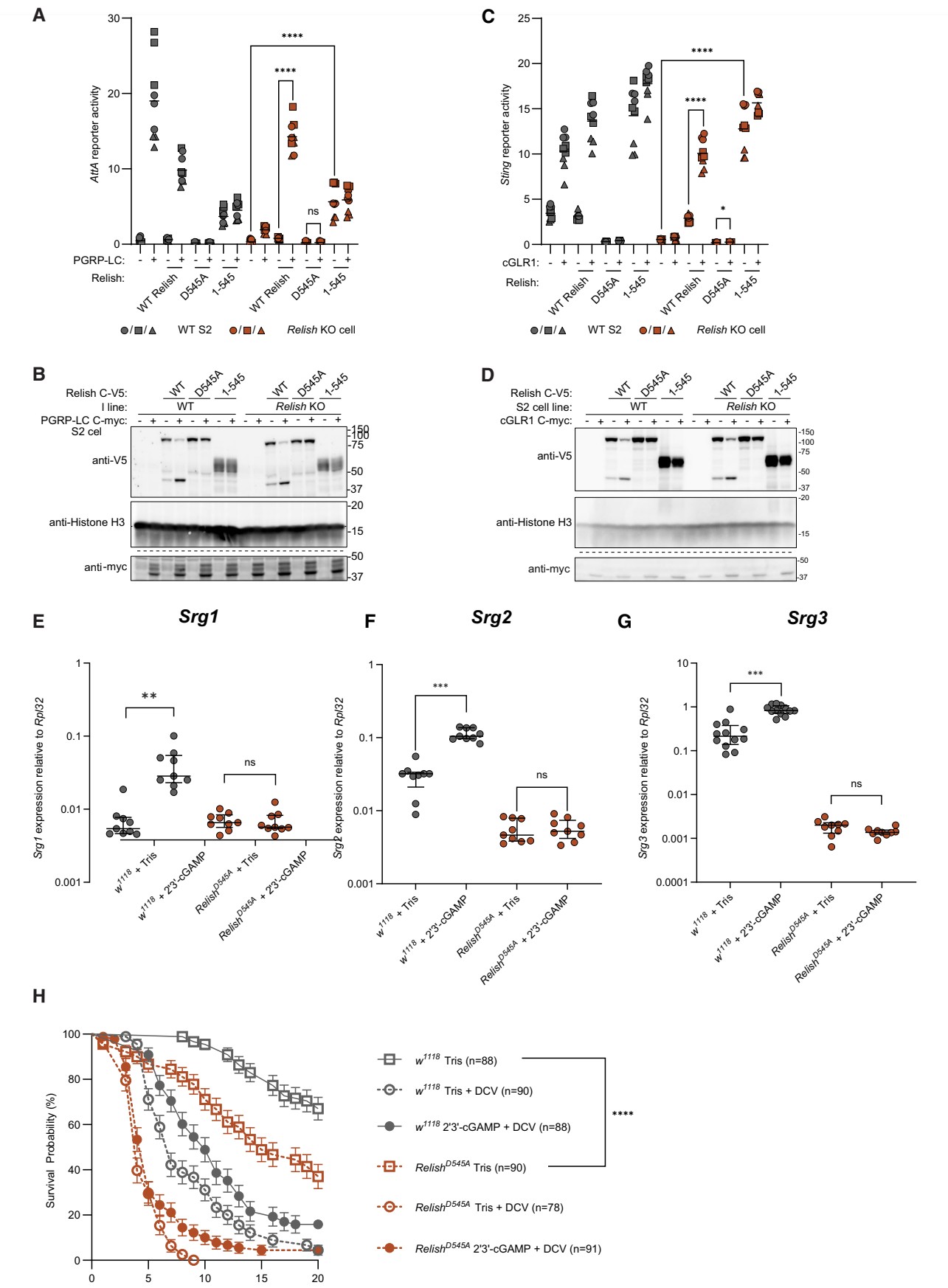

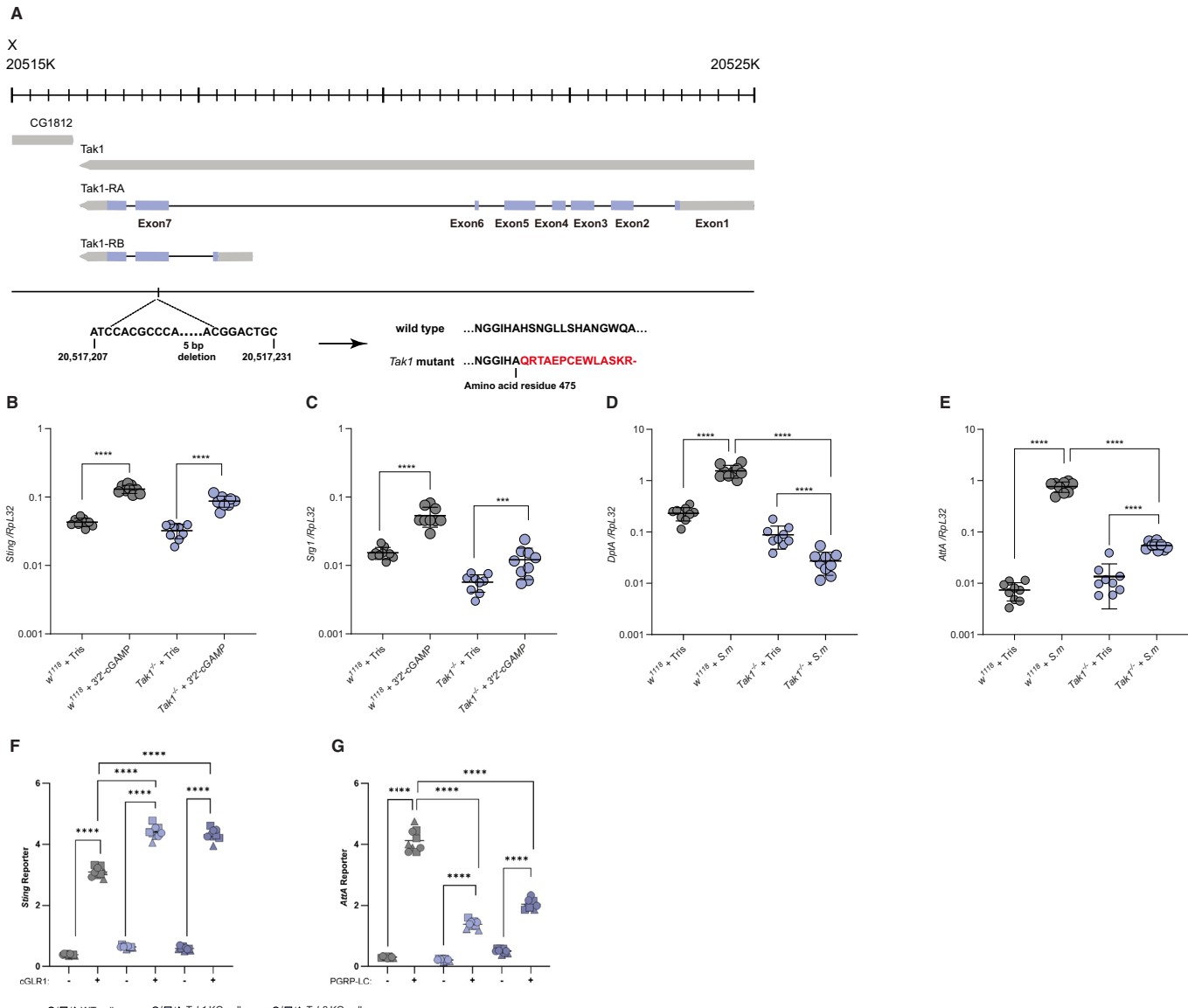

**Figure EV2. The role of dTAK1 and dTAB2 in dSTING and IMD signaling.**

(**A**) Generation of *Tak1* knock-out flies. The *Tak1* gene, located on the right arm of the X chromosome, is shown together with its annotated transcripts, Tak1-RA and Tak1-RB, encoding the long or short isoform of dTak1, respectively. Open reading frames are indicated in light purple. A 5 bp deletion creates a frameshift after the alanine residue at position 475 of the long isoform, leading to termination of translation after insertion of a 14 amino acid insertion (Gln-Arg-Thr-Ala-Glu-Pro-Cys-Glu-Trp-Leu-Ala-Ser-Lys-Arg). (**B–E**) Induction of the dSTING-induced genes *Sting* and *Srg1* (**B, C**) or IMD-induced genes *DptA* or *AttA* (**D, E**) in *w^1118* (control) or *Tak1^−/−* flies, measured by qPCR 24 h after intrathoracic injection of the STING agonist 3′2′-cGAMP or the IMD activator *Serratia marcescens* (*S.m*), respectively. Each data point is derived from a pool of six flies (three male, three female). Bars represent mean ± standard deviation. *P* values were calculated using one-way ANOVA with Tukey's multiple comparisons test: ****$p < 0.0001$, ***$p = 0.0001$. (**F, G**) Induction of the *Sting* or *AttA* reporter in WT (gray), *Tak1* (light purple) or *Tab2* (dark purple) KO S2 cells upon co-expression of cGLR1 or PGRP, respectively. Data from three independent experiments (different geometrical icons), each performed in biological triplicate, are shown with mean ($n = 9$). *P* values were calculated using two-way ANOVA, corrected with Tukey's post hoc test: ****$p < 0.0001$.

a

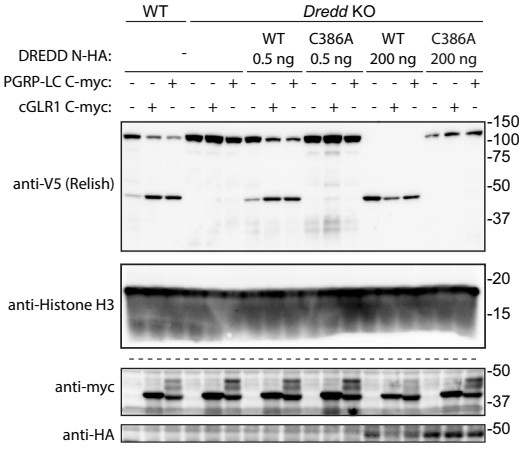

b **Srg1**

c **Srg2**

d **Srg3**

e **Srg1**

f **Srg2**

g **Srg3**

◀ **Figure EV3. DREDD and dFADD are necessary for signaling.**

(**A**) Cleavage of ectopically expressed Relish C-V5 in WT or *Dredd* KO S2 cells in response to expression of cGLR1 or PGRP-LC. *Dredd* KO cells were reconstituted with either WT DREDD or a catalytically inactive mutant (C386A) at either lower (0.5 ng plasmid) or higher (200 ng plasmid) expression levels. Main Fig. 2a shows a sub-portion of data from this figure. Lysates run on separate gels are indicated by dotted lines. (**B–G**), Induction of *Srg1, Srg2, or Srg3* in *yw* (control) and *yw*$^{Dredd-/-}$ flies (**B–D**) or *w*$^{1118}$ (control) and *Fadd*$^{-/-}$ flies (**E–G**) measured by qPCR 24 h after intrathoracic injection of the STING agonist 3'2'-cGAMP. Each data point is derived from a pool of six flies (three male, three female). Bars represent mean ± standard deviation. *P* values were calculated using a pairwise permutation test corrected with the Benjamini–Hochberg method: ***$p = 0.000734$, **$p = 0.001001$ (**B**, *yw*), 0.00545 (**B**, Dredd$^{-/-}$), or 0.005422 (**E**), *$p = 0.01087$ (**D**), 0.04722 (**E**), 0.01735 (**F**), 0.03859 (**G**, *w*$^{1118}$), or 0.02056 (**G**, Fadd$^{-/-}$), ns: $p = 0.1291$ (**C**), 0.5239 (**D**), or 0.3662 (**F**). Source data are available online for this figure.

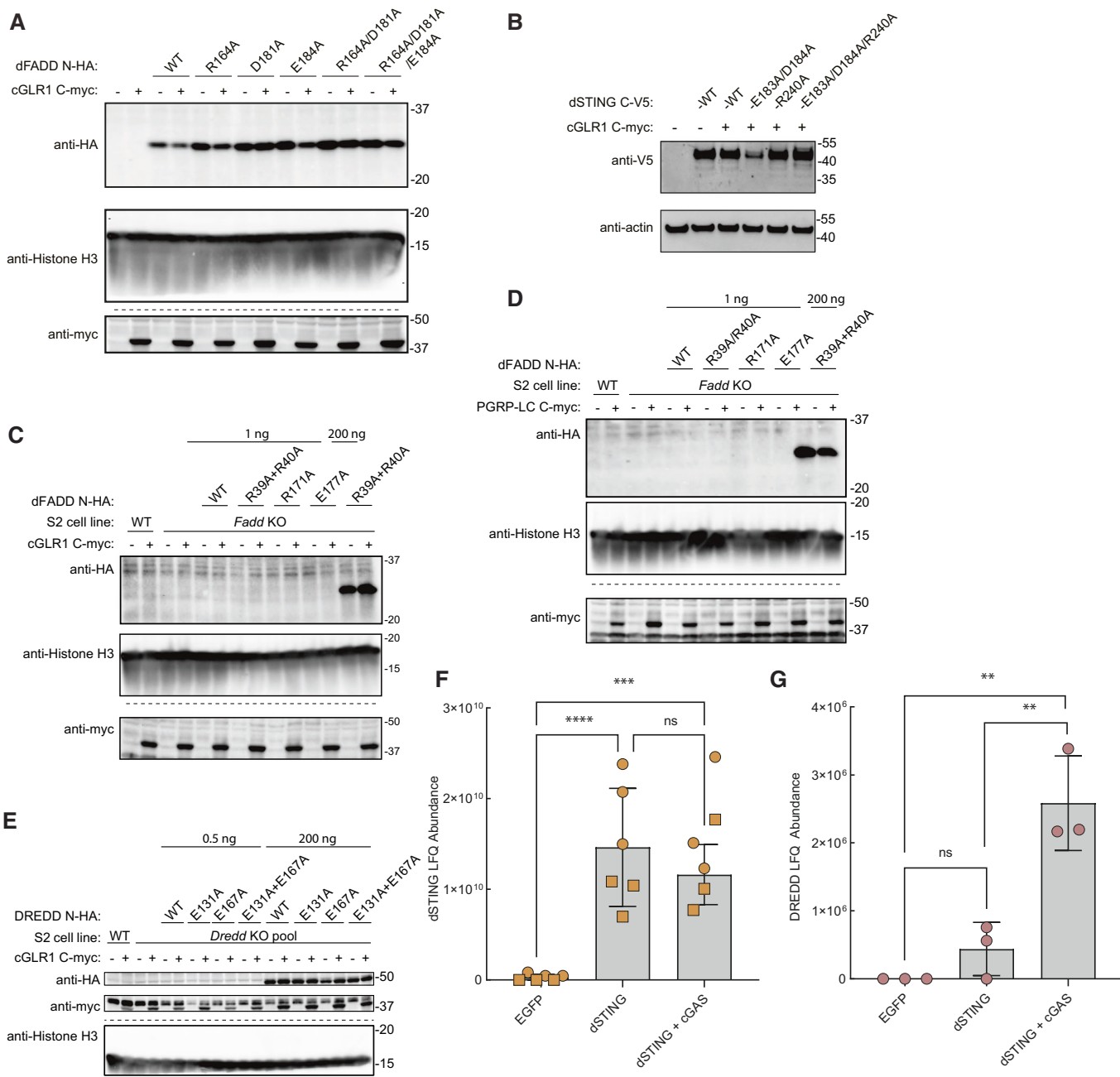

**Figure EV4.  Expression of dFADD, dSTING and DREDD mutants.**

(A) Western blot showing expression of dFADD mutants at high expression levels (200 ng plasmid) to enable detection by immunoblot in contrast to expression levels rescuing signaling in *Fadd* KO cells (0.5 ng expression plasmid). Lysates run on separate gels are indicated by dotted lines. (B) Western blot showing expression of dSTING mutants at high expression levels (750 ng plasmid) to enable detection by immunoblot in contrast to expression levels rescuing signaling in *Sting* KO cells (10 ng plasmid). (C–E) Western blot showing expression of dFADD and DREDD mutants. Mutants were additionally expressed with 200 ng plasmid to verify protein expression. Lysates run on separate gels are indicated by dotted lines. (F, G) LC/MS -based detection of dSTING (F) or DREDD (G) peptides from co-immunoprecipitation of ectopically expressed EGFP (negative control) or V5-tagged dSTING in WT S2 cells with or without co-expression of cGAS. In (F), data from $n = 2$ independent experiments each containing three biological replicates are shown. In (G) data from $n = 1$ independent experiment containing three biological replicates are shown, since unique peptides from DREDD were not confidently detected in the other experiment. Bars indicate the mean. $p$ values were calculated using one-way ANOVA corrected with Tukey's post hoc test: ****$p < 0.0001$, ***$p = 0.0009$, **$p = 0.0011$ (EGFP vs dSTING+cGAS) or 0.0031 (dSTING vs dSTING+cGAS), ns: $p = 0.4537$ (F) or 0.5118 (G). Source data are available online for this figure.

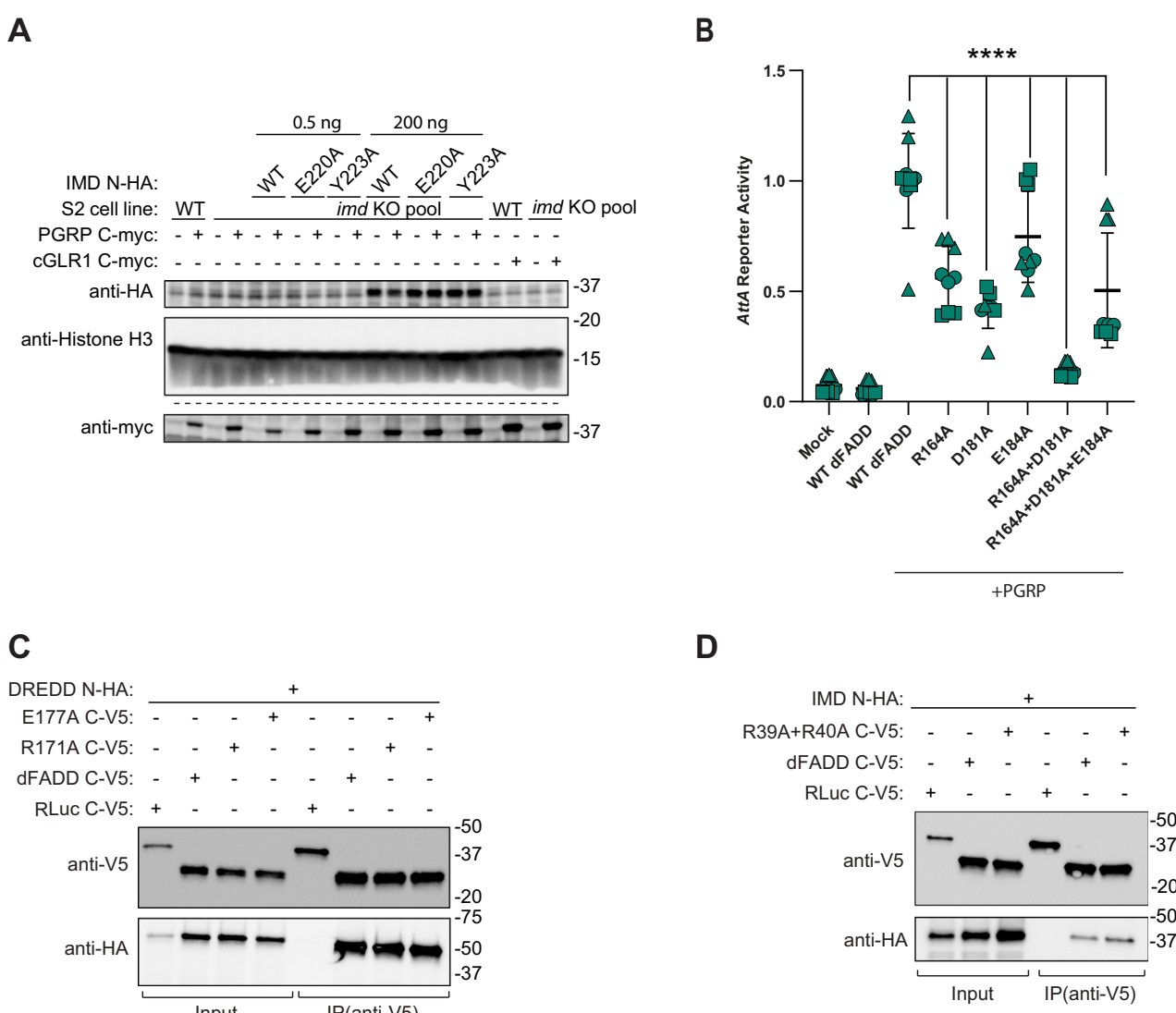

Figure EV5. Effect of mutations on IMD interactions and signaling.

(A) Western blot showing expression of IMD mutants. Mutants were additionally expressed with 200 ng plasmid to verify protein expression. Lysates run on separate gels are indicated by dotted lines ($n = 1$). (B) Induction of the *AttA* reporter in *Fadd* KO S2 cells upon co-expression of PGRP and reconstitution with WT dFADD or mutants disrupting the predicted dSTING:dFADD interface. Data from three independent experiments (different geometrical icons), each performed in biological triplicate ($n = 9$), are shown with mean and bars indicating standard deviation. For each experiment, all measurements were normalized to the mean of WT dFADD + PGRP. *P* values were calculated using two-way ANOVA, corrected with Dunnett's post hoc test: ****$p < 0.0001$. (C) DREDD N-HA and dFADD C-V5 or mutants disrupting the dFADD:IMD interface were expressed in S2 cells and immunoprecipitated on anti-V5 beads ($n = 2$). (D) IMD N-HA and dFADD C-V5 or the R39A/R40A mutant were expressed in S2 cells and immunoprecipitated on anti-V5 beads ($n = 2$). Source data are available online for this figure.

