## [Peer Review File · The EMBO Journal]

FADD is recruited to activated STING oligomers to initiate caspase-mediated NF- κ B activation in *Drosophila melanogaster*

Kasper Winther, Juliette Schneider, Gabrielle Haas, Anna Christensen, Shreya Goyal, Weisheng Luo, Ziming Wei, Jiyong Lui, Katarzyna Kjøge, Jan Enghild, Carine Meignin, Neal Silverman, Hua Cai, Jean-Luc Imler, and Rune Hartmann

Corresponding author(s): Rune Hartmann (rh@mbg.au.dk)

Review Timeline:

Submission Date:	4th Aug 25
Editorial Decision:	28th Sep 25
Revision Received:	17th Dec 25
Editorial Decision:	3rd Feb 26
Revision Received:	11th Mar 26
Accepted:	11th Mar 26

Editor: Ioannis Papaioannou

Transaction Report:

Dear Rune,

Thank you again for the submission of your manuscript EMBOJ-2025-121985 for consideration by The EMBO Journal, and for your patience during peer review. As I have already informed you, your manuscript has now been seen by three experts in the field, and we have received their detailed, informative, and constructive reports, which are included below.

I am very pleased to say that the referees recognize that this is an important, relevant, and timely topic, indicate interest in your findings, find your study well-designed, performed, and presented, and expect your manuscript to be of interest to the field. They also identify, however, a number of limitations, and they make several suggestions for the strengthening of the work and the increase of its impact on the field.

In light of the largely positive referees' comments and recommendations, I would like to invite you to submit a revised version of your manuscript taking the referees' suggestions on board, along with a detailed point-by-point response addressing all referees' comments. Please note that it is The EMBO Journal policy to allow only a single round of major revision, and acceptance of your manuscript will therefore depend on the completeness of your responses in this revised version.

Please let me know if you have any questions or comments that you would like to discuss with me; given the number and nature of the comments and suggestions made by the referees, I think it would be useful to discuss your revision plan already during the initial phase of your revisions. You are very welcome to share with me a draft point-by-point response letter/revision plan explaining if there are any points you do not agree with or cannot address, or alternatively we could arrange a video call, if you prefer.

We generally allow three months as standard revision time (December 27, 2025). As a matter of policy, competing manuscripts published during this period will not negatively impact our assessment of the conceptual advance presented by your study. However, we request that you contact us as soon as possible upon publication of any related work, to discuss how to proceed. Should you foresee a problem in meeting this three-month deadline, please let us know in advance and we will be able to grant an extension.

Thank you for the opportunity to consider your work for publication in The EMBO Journal. I look forward to your revision.

Best regards,

Ioannis

Instructions for preparing your revised manuscript

1. When you are ready to submit the revision, please upload:

- A Word file of the manuscript text (including legends of main Figures, EV Figures and Tables). Please make sure that changes are highlighted (or "tracked") to be clearly visible.

- Individual production-quality figure files (one file per figure). When assembling your figures, please refer to our figure preparation guidelines in order to ensure proper formatting and readability in print as well as on screen:

If the data shown in a figure are obtained from n {less than or equal to} 2, please use scatter plots showing the individual data points.

- i. the name of the statistical test used to generate error bars and P values
- ii. the number (n) of independent experiments (please specify technical or biological replicates) underlying each data point (discussion of statistical methodology can be reported in the Materials and Methods section, but figure legends should contain a basic description of n , P , and the test applied)
- iii. the nature of the bars and error bars (s.d., s.e.m.).

- A point-by-point response to the referees' comments, with a detailed description of the changes made (as a word file). All referees' concerns must be fully addressed and their suggestions taken on board. When preparing your letter of response to the referees' comments, please bear in mind that this will form part of the Review Process File and will therefore be available online to the community. Please note that you have the possibility to opt out of the transparent process at any stage prior to publication by letting the editorial office know (contact@embojournal.org); if you do opt out, the Review Process File link will point to the following statement: "No Review Process File is available with this article, as the authors have chosen not to make the review process public in this case.". For more details on our Transparent Editorial Process, please visit our website: <https://www.embopress.org/page/journal/14602075/authorguide#transparentprocess>

- Expanded View (EV) files (replacing Supplementary Information) that are collapsible/expandable online. A maximum of 5 EV Figures can be typeset. EV Figures should be cited as "Figure EV1, Figure EV2" etc. in the text, and their respective legends should be included in the manuscript file after the legends of regular figures. See detailed instructions regarding Expanded View files here: <https://www.embopress.org/page/journal/14602075/authorguide#expandedview>

- For the figures that you do NOT wish to display as Expanded View figures, they should be bundled together with their legends in a single PDF file called "Appendix", which should start with a short Table of Contents (including page numbers). Appendix figures should be referred to in the main text as: "Appendix Figure S1, Appendix Figure S2" etc. Please see detailed instructions here: <https://www.embopress.org/page/journal/14602075/authorguide#expandedview>

- A complete author checklist, which you can download from our author guidelines (<https://www.embopress.org/page/journal/14602075/authorguide>). Please note that the checklist will also be part of the Review Process File.

2. Please note that no statistics should be calculated and shown in Figures if $n=2$. Please also note that each p value should be reported as an exact value.

3. Before submitting your revision, primary datasets (and computer code, where appropriate) produced in this study need to be deposited in appropriate public databases (see <https://www.embopress.org/page/journal/14602075/authorguide#dataavailability>). In particular, the mass spectrometry data generated in your study must be deposited in an appropriate repository. The accession numbers, database, and the specific URLs (links) should be listed in a formal "Data availability" section (placed after Methods), following the example below:

"The RNA-seq datasets produced in this study are available in the following database:
Gene Expression Omnibus GSE46843 (<https://www.ncbi.nlm.nih.gov/geo/query/acc.cgi?acc=GSE46843>)"

*** All links should resolve to a page where the data can be accessed. ***

*** Please remember to provide in the Data availability section of your revised manuscript reviewer passwords if the datasets are not yet public. ***

*** The Data Availability Section is restricted to new primary data that are part of this study. In case you have no data that require deposition in a public database, please state so instead of referring to the database: "Our study includes no data deposited in public repositories." under the heading "Data availability". ***

4. The materials and methods need to be described in the manuscript using our structured methods format, which is now required for all research articles. According to this format, the Methods section includes a single "Reagents and Tools Table" - listing key reagents, experimental models, software and relevant equipment including their sources and relevant identifiers - followed by a "Methods and Protocols" section describing the methods. Please download and fill our Reagents and Tools Table template (.docx), which you can find in our author guide: <https://www.embopress.org/page/journal/14602075/authorguide#structuredmethods>. When submitting your revised manuscript, please do not include the Reagents and Tools Table in the Methods section of the manuscript but instead upload it as a separate file choosing the file type "Reagent Table".

5. Please check that the title and the abstract of the manuscript are brief, yet explicit, even to non-specialists. The length of the title should not exceed 100 characters, and the abstract should be a single paragraph not exceeding 175 words.

6. Please also note our reference format: <https://www.embopress.org/page/journal/14602075/authorguide#referencesformat>.

8. Please remember: digital image enhancement is acceptable practice, as long as it accurately represents the original data and conforms to community standards. If a figure has been subjected to significant electronic manipulation, this must be noted in the figure legend or in the "Materials and Methods" section. The editors reserve the right to request original versions of figures and the original images that were used to assemble the figure.

9. Our journal encourages inclusion of data citations in the reference list to directly cite datasets that were obtained from public databases. Data citations in the article text are distinct from normal bibliographical citations and should directly link to the database records from which the data can be accessed. In the main text, data citations are formatted as follows: "Data ref: Smith et al, 2001" or "Data ref: NCBI Sequence Read Archive PRJNA342805, 2017". In the Reference list, data citations must be labeled with "[DATASET]". A data reference must provide the database name, accession number/identifiers, and a resolvable link to the landing page from which the data can be accessed at the end of the reference. Further instructions are available at: <https://www.embopress.org/page/journal/14602075/authorguide#referencesformat>.

10. We request authors to consider both actual and perceived competing interests. Please review our policy (<https://www.embopress.org/page/journal/14602075/authorguide#conflictofinterest>) and update your competing interests statement if necessary. Please name this section 'Disclosure and competing interests statement' and place it after the Acknowledgements section.

11. Please note that all corresponding authors are required to provide an ORCID ID upon submission of a revised manuscript (<https://orcid.org/>). Please find instructions on how to link your ORCID ID to your account in our manuscript tracking system in our Author guidelines (<https://www.embopress.org/page/journal/14602075/authorguide#authorshipguidelines>).

12. We use CRediT to specify the contributions of each author in the journal submission system. CRediT replaces the author contribution section, which should be removed from the manuscript. Please use the free text box to provide more detailed descriptions. See also guide to authors: <https://www.embopress.org/page/journal/14602075/authorguide#authorshipguidelines>.

14. We would also welcome the submission of cover suggestions or motifs to be used by our Graphics Illustrator in designing a cover.

15. Please use the link below to submit your revision:
<https://emboj.msubmit.net/cgi-bin/main.plex>

Referee #1:

EMBOJ-2025-121985

FADD is recruited to activated STING oligomers initiating caspase-mediated NF- κ B activation in *Drosophila melanogaster* by Kasper G Winther et al

It has previously been reported that *Drosophila* cGAS-like receptors activate antiviral responses by activating Sting, which in turn induces expression of Sting regulated genes via the NF- κ B Relish. In this manuscript, the authors have studied the mechanism of Sting-induced Relish activation, comparing it to the related Imd signaling mechanisms with the aim to elucidate the differences in activation of the two pathways. The authors show that Relish cleavage is required for activation of the Sting pathway, similarly as for Imd signaling. Using reporters, they also verify that Sting-dependent gene expression requires Sting and not Imd, while Imd-dependent gene expression requires Imd but not Sting, verifying that the pathways can be induced independent of one another. Interestingly, the authors show that Fadd and Dredd, which are known to mediate Imd signaling, also are needed for Sting-activation, while Tak1/Tab2 are needed only for Imd signaling. Further, the authors studied the Sting-Fadd and the Fadd-Dredd interactions by AlphaFold modelling and immunoprecipitation. They suggest that oligomerization of Sting dimers to tetramers during activation leads to formation of a binding surface that recruits Fadd. By expressing point mutants in S2 cells, they show that the ability of Sting to bind Fadd is required for cGLR1-induced Sting activation, but not for Imd activation. Meanwhile, they show that the Fadd-Dredd interaction is needed both for Dredd and Fadd to induce Imd and Sting reporters.

This is a thoroughly conducted study that addresses an important and timely question. Sting-induced NF- κ B signaling has not been well described before, and this work provides significant mechanistic insight into previously suggested responses. Because of its robust genetic tools and the conservation of key signaling molecules, *Drosophila* is well-suited for studying molecular

mechanisms of Sting and NF- κ B signaling. The methods used are well suited to address the research questions, and the experimental work is of high quality, yielding convincing results. The findings advance our understanding of the molecular mechanisms regulating antiviral responses, making the study highly relevant and conceptually interesting. Importantly, its implications extend beyond the *Drosophila* community, offering insights of broad interest to the wider scientific field. The following comments are given as suggestions to support the conclusions and further improve the manuscript.

Major concerns

1. The AlphaFold model suggests that oligomerization of Sting dimers is needed for Fadd to bind. However, the data from experiments using point mutations suggests that binding to only one of the dimers is enough to induce Sting signaling. Possible causes for the result are discussed in the manuscript, but it is not tested if oligomerization to tetramers is required for functional signaling. Hence, it should be experimentally tested if Fadd can bind Sting mutants that cannot form tetramers, and if Relish cleavage and Sting signaling can be induced by such mutations.
2. The authors have performed LC/MS to study interactions of Fadd upon Sting activation. However, it is not mentioned if Sting and Dredd peptides were found in the screen. If this is not the case, the figure does not seem logic to include.
3. The immunofluorescence experiment is not convincingly presented. There is no visual sign of co-localization in the microscopy images shown. Instead, in the separate panels, Sting seems to be similarly localized in all cells but expressed a bit more in the sample where colocalization is suggested. Meanwhile, Fadd seems to be similarly localized in all cells except when co-expressed with mutant STING and treated with cGLR1. This is however not reflected in the statistics. In addition, the description of the experiment is unclear and no controls to exclude unspecific staining are included. Finally, the marking of the graph in figure 3g with two overlapping lines for cGLR1 seems to be wrong.
4. The AlphaFold model indicates that the Fadd-Dredd interaction is mediated via E131 and E167 of Dredd. Of these residues, mutation of E167 seems to be important for Sting and Imd signaling, while mutation of E131 does not seem to have an effect. Is mutation of E131A not sufficient to interfere with binding to Fadd?
5. It is shown that Sting signaling is abrogated by loss of the Sting-Fadd and the Fadd-Dredd interaction. Is also cGLR1-induced Relish cleavage affected in the absence of these interactions?
6. Dredd is autoprocessed by itself upon its activation. Hence, the cleaved forms should also be visible at the western blots after activation.
7. It would be important to verify that the Sting E183A/D184A/R240 mutant that does not bind Fadd does not affect Imd signaling and conversely that the Imd E220A/Y223A mutant that does not bind Fadd does not affect Sting signaling. This would show that Sting and Imd induces Fadd and Dredd activated Relish cleavage and activation separately from one another.
8. All experiments must be repeated at least three times, all quantitative data should include statistics (including the expanded view contents), and the information about the repetitions need to be clearly explained. In the current manuscript, it is not clear how the survival experiments are repeated. Large n-numbers are indicated in the figures, but are these from different experiments or from one?
9. The Sting and AttA reporters are induced strongly by cGLR1 and PGRP-LC (respectively) in WT cells in figures 1d-g, ED2a-b and ED4f-g, but not at all in figures 4b,c,e,f and 5b,d,f. What is the difference in experimental setup?

Minor concerns

10. Mammalian Sting signaling is discussed in the manuscript, but it should also be mentioned if the Sting, Fadd and caspase interaction surfaces are conserved in mammals and whether they are involved in NF- κ B signaling induced via mammalian Sting.
11. The rationale behind using injection of 2'3'-cGAMP/3'2'-cGAMP to activate the Sting pathway to protect from DCV infection, which should induce the same pathway is not explained.
12. In the manuscript, it is written that the response of Tak1^{-/-} flies to Gram negative bacteria (ED 4e) was highly impaired although the fold induction of AttA is significantly induced (with ****). It would be more correct to say that it is reduced compared to control.
13. In figure 2 and 3, in the figure markings and figure legends all transfected plasmids are not indicated (Relish-V5 and Sting-HA).
14. The correct nomenclature for fly proteins and genes should be used (as described in FlyBase).

Referee #2:

The report by Winther et al. addresses the molecular mechanisms related to the activation of cGAS/STING signalling in *Drosophila melanogaster* and, in particular, how STING activates the NF- κ B family transcription factor, Relish. The manuscript details the authors' efforts in delineating the steps required for Relish activation downstream of crucial antiviral effectors involved in STING signalling. The authors use a combination of in vitro CRISPR efforts in *Drosophila* S2 cells combined with in vivo experiments in flies to suggest that FADD and DREDD are crucial for STING-dependent induction of Relish cleavage and downstream expression of STING target genes. The authors also use structural biology predictions to suggest that FADD directly binds to STING oligomers and that this binding is important for the response to viral infection. Moreover, again using structural biology predictions, the authors describe the potential molecular mechanisms by which FADD interacts with DREDD and IMD and provide a potential explanation for the fact that FADD and DREDD, but not IMD, are involved in both STING antiviral signalling and IMD anti-bacterial signalling.

In general, experiments and data are well presented and appropriately controlled and the data is of high quality. The authors provide extensive extended data to complement the main figures and make a compelling case for their proposed model of how STING signalling is regulated. The manuscript should be of great interest for researchers working on mechanisms regulating the response to viral infection and for the larger innate immunity field.

Comments on the manuscript are shown below.

Major points:

Regarding Extended Data Figure 1, authors should comment on why the Relish KO cells appear to have a higher PCR amplicon than the control WT cells? Do the cells have any duplication of sequences to account for this observation?

Authors should comment on why the non-cleavable Relish mutant (D545A) is acting as a dominant negative allele. Also, authors should comment on why the C-terminally truncated form of Relish (1-545) seems to be much more efficient in activating the STING reporter than the IMD reporter at basal levels without stimulation (Extended Data Figure 2a,c).

One important point that the authors do not explore is whether there is any synergy, competition or cross-activation between the STING and IMD pathways. In nature, it is conceivable that injury would result in simultaneous infection with bacteria and viruses. How would this be tackled by the immune system? Does PGRP-LC lead to activation of the STING reporter? Does cGLR1/2 activate the IMD reporter? Is there any indication that there is a crosstalk between the two pathways? The fact that both share components could indicate that this is a possibility.

Authors should discuss why 3'2'-cGAMP was used in the experiments with the Tak1 mutant and not in others. Is there a chance that some phenotypes are dampened due to the use of 2'3'-cGAMP?

Authors should clarify or test whether STING overexpression is sufficient to activate the STING reporter in dFADD KO cells. This would strengthen the argument that dFADD is essential for STING signalling.

dFADD re-expression in dFADD KO cells is not shown in WB experiments.

Authors should comment on why the baseline of Srg gene expression is so low in FADD mutants.

Authors should test whether the dFADD mutations created to abrogate binding to STING affect IMD signalling. If so, this could reveal structural changes in dFADD beyond simply affecting interaction with STING.

Related to the previous point, authors should comment on whether the E183A and R240A mutations in STING affect STING dimerisation. Additionally, authors could test whether dFADD can co-IP a STING mutant that cannot dimerise. According to the authors' proposed model, dFADD should not meaningfully interact with a monomeric STING molecule.

If possible, authors should provide evidence that the dIKK β -dIKK γ complex is not required for STING-induced Relish cleavage and downstream induction of STING target genes to complement the analysis of the dTak1-dTab2 complex.

According to Ertürk-Hasdemir et al. (2009), it should be possible to detect phosphorylated Relish in the S2 system (S528/S529). Is Relish phosphorylation similarly affected by activation of the IMD pathway or the STING pathway? Based on the authors' observations and the lack of effect of dTak1, one would potentially predict that although both lead to Relish cleavage, activation of STING signalling may not lead to Relish phosphorylation (or at least not to the same extent). Authors should therefore discuss the importance/relevance of Relish phosphorylation for Relish cleavage in the context of STING signalling.

Authors should investigate downstream Relish cleavage in S2 cells when the dFADD mutations R164A, D181A and E184A are present (alone or in combination). Similarly, the effect of the mutations that affect the dFADD-DREDD interaction (dFADD R39A/R40A and DREDD E131A/E167A) on Relish cleavage should also be monitored.

Authors should comment on whether the dFADD-STING interaction is compatible with the dFADD-DREDD interaction or whether they are mutually exclusive as that has relevance for their proposed model.

Regarding Extended Data Figure 10d, there is a potential issue with the prediction of dFADD-IMD interaction as most of the IMD molecule is a very low confidence prediction. How could this influence the proposed mechanism?

Minor points:

Related to Figure 3a, there is a mistake in the nomenclature of the dSTING mutation and aa residue. Panel 3a mentions D183 but panel 3c and the main text mentions E183. This should be corrected in the final version of the manuscript.

Related to Figure 1, authors should clarify which PGRP-LC isoform(s) are being expressed in S2 cells.

Authors observe that RelishD545A mutant flies are more susceptible to DCV infection. Is this also the case if flies are raised in axenic conditions?

Protection by 2'-3'-cGAMP injection does not appear to be very extensive even when flies are WT. Why is this? Does expression of cGRL1 lead to a similar level of rescue or a more effective rescue? Are there other, more important antiviral pathways at play?

Related to Figure 3f,g and the co-localisation of dFADD and STING in S2 cells, authors should perhaps increase the number of cells analysed given that dFADD and STING appear to have large areas where they do not colocalise.

The inclusion in the Introduction of a discussion on the production of CDNs in bacteria is perhaps not necessary to provide a relevant background for the experimental work performed.

In most of the Extended Data Figures, the expressed constructs are not detectable in WB when at low concentration and, therefore, do not really serve as controls.

Histone H3 loading control blots could have been optimised in some of the figure panels as they seem overexposed making it more challenging to clearly determine equal protein loading in all samples.

Methods section is thorough, with only minor omissions (cell number used in WB experiments, transfection reagent used in dSTING IP and MS experiments).

Minor typos present (particularly in Methods section).

Referee #3:

Referee Report on EMBO J Submission: "FADD is recruited to activated STING oligomers initiating caspase-mediated NF- κ B activation in *Drosophila melanogaster*"

This is an interesting and timely study that addresses an important mechanistic question in the field of STING signaling. The authors provide convincing evidence that FADD is recruited to oligomerized STING complexes in *Drosophila* and functions downstream to activate the NF- κ B-like transcription factor Relish. The structural modeling of FADD interaction with STING oligomers, together with functional data, advances our understanding of how the conserved FADD/caspase axis is integrated into distinct immune pathways. The work is carefully executed, the conclusions are significant, and the manuscript is clearly written. Overall, I find the study appropriate for publication in The EMBO Journal.

That said, I have one major suggestion that I believe would substantially strengthen the manuscript. The central mechanistic claim is that FADD is recruited to STING via a specific interface that the authors have modeled in detail. While the biochemical and structural analyses are compelling, the model would be much more convincing if supported by genetic data *in vivo*. In particular, the authors could generate dFADD variants defective in STING binding (as defined by their structural predictions) and introduce these variants into the dFADD locus. Testing such flies for (i) susceptibility to viral infection and (ii) partial protection upon 3'-2'-cGAMP injection (analogous to the experiments presented in Fig. 1c with Relish D545A flies) would provide powerful validation of the model. Even partial data in this direction would significantly strengthen the manuscript. Apart from this, I have no major concerns.

Minor points could be addressed during revision, the authors may wish to expand briefly in the discussion on how the FADD-STING interaction compares to vertebrate STING signaling, which could help highlight the evolutionary insights.

Dear Dr. Papaioannou

We have now carefully read the review of our paper and are pleased. The reviewers have been very thorough and although they raise several points, we believe they are in general quite positive. Below you will find our detailed rebuttal letter, where we have done our utmost to respond in a similar thorough manner.

Looking forward to hearing your decision.

On behalf of the Authors

Rune Hartmann

Referee #1:

EMBOJ-2025-121985

FADD is recruited to activated STING oligomers initiating caspase-mediated NF- κ B activation in *Drosophila melanogaster*
by Kasper G Winther et al

It has previously been reported that *Drosophila* cGAS-like receptors activate antiviral responses by activating Sting, which in turn induces expression of Sting regulated genes via the NF- κ B Relish. In this manuscript, the authors have studied the mechanism of Sting-induced Relish activation, comparing it to the related Imd signaling mechanisms with the aim to elucidate the differences in activation of the two pathways. The authors show that Relish cleavage is required for activation of the Sting pathway, similarly as for Imd signaling. Using reporters, they also verify that Sting-dependent gene expression requires Sting and not Imd, while Imd-dependent gene expression requires Imd but not Sting, verifying that the pathways can be induced independent of one another. Interestingly, the authors show that Fadd and Dredd, which are known to mediate Imd signaling, also are needed for Sting-activation, while Tak1/Tab2 are needed only for Imd signaling. Further, the authors studied the Sting-Fadd and the Fadd-Dredd interactions by AlphaFold modelling and immunoprecipitation. They suggest that oligomerization of Sting dimers to tetramers during activation leads to formation of a binding surface that recruits Fadd. By expressing point mutants in S2 cells, they show that the ability of Sting to bind Fadd is required for cGLR1-induced Sting activation, but not for Imd activation. Meanwhile, they show that the Fadd-

Dredd interaction is needed both for Dredd and Fadd to induce Imd and Sting reporters.

This is a thoroughly conducted study that addresses an important and timely question. Sting-induced NF- κ B signaling has not been well described before, and this work provides significant mechanistic insight into previously suggested responses. Because of its robust genetic tools and the conservation of key signaling molecules, *Drosophila* is well-suited for studying molecular mechanisms of Sting and NF- κ B signaling. The methods used are well suited to address the research questions, and the experimental work is of high quality, yielding convincing results. The findings advance our understanding of the molecular mechanisms regulating antiviral responses, making the study highly relevant and conceptually interesting. Importantly, its implications extend beyond the *Drosophila* community, offering insights of broad interest to the wider scientific field. The following comments are given as suggestions to support the conclusions and further improve the manuscript.

First of all, we would like to thank the reviewer for a thorough and positive review and will do our best to address the concerns raised in the following.

Major concerns

1. The AlphaFold model suggests that oligomerization of Sting dimers is needed for Fadd to bind. However, the data from experiments using point mutations suggests that binding to only one of the dimers is enough to induce Sting signaling. Possible causes for the result are discussed in the manuscript, but it is not tested if oligomerization to tetramers is required for functional signaling. Hence, it should be experimentally tested if Fadd can bind Sting mutants that cannot form tetramers, and if Relish cleavage and Sting signaling can be induced by such mutations.

This question is relevant but very difficult to address experimentally in the *Drosophila* system. The formation of oligomeric STING is well established in vertebrates, both through structural studies using cryo-EM and through traditionally light microscopy^{1,2}. The Ablasser laboratory developed an immunofluorescence-based method to detect STING oligomerisation in human HEK293T cells, where STING is fused to a fluorescent protein and oligomerisation is observed as the formation of a punctate staining after activation of STING by co-transfection with cGAS. We already attempted this methodology in S2 cells, but the small size of the cytoplasm makes it difficult to observe clear punctate formation. We also attempted a biochemical characterisation of the dSTING oligomerisation by native electrophoresis followed by western blotting; however, this was also not technical possible in the S2 cells.

Given the restraints listed above, producing and verifying dSTING mutants that will not oligomerise is difficult but we have convincingly shown the specificity of dFADD recruitment to activated dSTING: dFADD only immunoprecipitate with activated dSTING, and in the reverse immunoprecipitation, dFADD only pulls down dSTING if dSTING is activated(fig 3d and e,)

2. The authors have performed LC/MS to study interactions of Fadd upon Sting activation. However, it is not mentioned if Sting and Dredd peptides were found in the screen. If this is not the case, the figure does not seem logic to include.

We did indeed detect DREDD-derived peptides, however, borderline significance, yet activity dependent, and we show the raw data below. We also detected an abundant amount of dSTING-derived peptides. In some of our replicates, we only detected one unique peptide for DREDD, which is below our threshold for quantification. We suggest including the following sentence in the text *“Furthermore, DREDD peptides were also detected in the samples of activated dSTING, however, at borderline significances as this is a secondary interaction to dSTING.”*, but not to include any new data in the paper. Below is shown an experiment performed in triplicate where we did detect two unique peptides, in the second set of 3 replicate we only detected one unique peptide.

3. The immunofluorescence experiment is not convincingly presented. There is no visual sign of co-localization in the microscopy images shown. Instead, in the separate panels, Sting seems to be similarly localized in all cells but expressed a bit more in the sample where colocalization is suggested. Meanwhile, Fadd seems to be similarly localized in all cells except when co-expressed with mutant STING and treated with cGLR1. This is however not reflected in the statistics. In addition, the description of the experiment is unclear and no controls to exclude unspecific staining are included. Finally, the marking of the graph in figure 3g with two overlapping lines for cGLR1 seems to be wrong.

There was indeed a labelling mistake in panel g. cGLR1 is only present for the last two samples on the right. Regarding the colocalization in the images, we have performed three additional transfection experiments to increase the number of cells analyzed. We have also used a different confocal microscope (Leica LSM780) and modified the settings of the Zeiss Axiovert 1 spinning disk microscope to improve the resolution. The images presented below are the best that were

obtained, and we do not feel that they are more convincing than in the initial version. This is probably because dFADD has a diffused distribution in the cells we use (S2 cells are small and roundish with poor adherence to the support) and only a fraction of the molecules are recruited to dSTING upon activation. Nevertheless, quantification of Pearson's correlation ratio always indicates a statistically significant difference for the colocalization of dSTING and dFADD upon pathway activation by cGLR1, whether we use the in-built tool of Fiji (Coloc) or the JaCOP plug-in.

Because we cannot provide convincing representative images from cells, and these data were only used as confirmation of the thorough pull-down experiments we conducted, we believe it is best to remove these data from the manuscript. Thus, figure 3f and g have been removed as well as the associated text.

4. The AlphaFold model indicates that the Fadd-Dredd interaction is mediated via E131 and E167 of Dredd. Of these residues, mutation of E167 seems to be important for Sting and Imd signaling, while mutation of E131 does not seem to have an effect. Is mutation of E131A not sufficient to interfere with binding to Fadd?

First, a brief general description of our approach to verifying AlphaFold models by mutagenesis. We examine the AlphaFold prediction for side chain specific interactions, which we can target by

mutating a single amino acid, or in some cases a pair of amino acids, without introducing changes in the protein backbone or disordering the general fold. As interactions between macromolecules involve larger surface areas with multiple interactions, changing a single amino acid is not necessarily expected to have an all or nothing effect, but changing multiple amino acids is often expected to have a cumulative effect, if the original prediction is correct.

One can easily introduce mutations, like deletions, which disrupt or destabilise overall fold, but in our opinion, this just confirms general protein function and not the AlphaFold prediction. We have chosen the approach described above and systematically provide data for all the mutants we made to provide the reader with the full picture.

Regarding the specific question asked; yes, it would indeed seem that E131A alone is insufficient to disrupt the interaction, for the reasons explained above, yet for STING there is a small but clear additive effect of E131A when combined with E167A in terms of reporter activity. We have also verified this using immunoprecipitation, please see data below.

5. It is shown that Sting signalling is abrogated by loss of the Sting-Fadd and the Fadd-Dredd interaction. Is also cGLR1-induced Relish cleavage affected in the absence of these interactions?

We have showed that the absence of both dFADD and DREDD impairs dSTING-mediated cleavage of Relish and that the mutations in question abolish the physical interaction of dSTING with dFADD and dFADD with Dredd, respectively, as well as disrupt signalling. Furthermore, the catalytically inactive version of DREDD (C386A) does not rescue *Dredd* KO cells (Fig. 2a), neither does the non-cleavable version of Relish mediate signalling (D545A, Fig. 1). The cleavage assays in the KO cells

are time consuming and difficult due to the low transfectability of S2 cells, and we do not consider those experiments necessary in light of the delay of publication that this will cause.

6. Dredd is autoprocessed by itself upon its activation. Hence, the cleaved forms should also be visible at the western blots after activation.

We are only blotting for DREDD in Fig. 4d and g, where we are examining the dFADD:DREDD interaction, which is constitutive, hence we are not using cGLR1 for activation. To visualise DREDD cleavage under activating conditions would require an entirely new experimental set up, which might be relatively difficult as the fragment of DREDD is unstable and the timeframe of our transfection experiment is quite long. This was not a priority for us as interdomain cleavage of DREDD was shown to be neither sufficient nor necessary for IMD-mediated cleavage of Relish³.

7. It would be important to verify that the Sting E183A/D184A/R240 mutant that does not bind Fadd does not affect Imd signaling and conversely that the Imd E220A/Y223A mutant that does not bind Fadd does not affect Sting signaling. This would show that Sting and Imd induces Fadd and Dredd activated Relish cleavage and activation separately from one another.

We have performed the requested experiments, and the result are shown below. As shown in the left hand figure, mutating E220A alone or both E220A and Y223A in IMD does not impact dSTING signalling in *Imd* KO cells. Likewise, the dSTING E183A/D184A/R240 mutant, or the respective single mutants, have no or little effect upon IMD signalling (right hand figure).

B**C**
We have already shown that dSTING signalling is unaffected in *lmd* KO cells and *vice versa*. Therefore, we do not believe that mutating IMD, in a scenario where IMD is not required, would be productive, or *vice versa*. Hence, the suggested experiments are, in our opinion, redundant.

8. All experiments must be repeated at least three times, all quantitative data should include statistics (including the expanded view contents), and the information about the repetitions need to be clearly explained. In the current manuscript, it is not clear how the survival experiments are repeated. Large n-numbers are indicated in the figures, but are these from different experiments or from one?

All experiments were performed at least three times, unless specified otherwise (Fig. 2a, 3e, n=2). We realise that this is not ideal, however, the first author has already left the laboratory, and it will be time consuming to restart experiments. Furthermore, we have a series of internal controls or independent experiments that address the same issue to validate our data.

For Fig. 2a, this is a Relish cleave assay, where our claim is that activation of either the dSTING or IMD pathway leads to Relish cleavage. This is shown twice on each blot, in WT cells and in *Dredd* KO cells, which are rescued with ectopic DREDD expression. Thus, I will argue that each blot contains an internal duplication, and we have then repeated the entire experiment. Furthermore, for the

Relish cleavage, we performed a series of preliminary experiments to set up the optimal conditions, which gave identical results.

For Fig. 3e, which is an immunoprecipitation experiment where we show activated that dSTING co-immunoprecipitates with dFADD. Here we also only have two replicates, but Fig. 3d addresses the exact same question, just using mass spectrometry-based co-immunoprecipitation, which was performed in 6 replicates. In effect, Fig. 3d and f are duplications of the same experiment using different techniques, which we also clearly write in the text.

For *in vivo* survival experiments, we apologize for omitting to mention in the Methods section that two independent experiments were performed, each involving three biological replicates. This has been corrected.

9. The Sting and AttA reporters are induced strongly by cGLR1 and PGRP-LC (respectively) in WT cells in figures 1d-g, ED2a-b and ED4f-g, but not at all in figures 4b,c,e,f and 5b,d,f. What is the difference in experimental setup?

Fig. 4b and c are *Fadd* KO cells and Fig. 4e and f are *Dredd* KO cells, hence activation occurs only when the activator (cGLR1 or PGRP) is expressed simultaneously with a rescue plasmid expressing exogenous DREDD or dFADD. All cells are KO, and the WT in the figure legend refers to if the rescue was performed with a plasmid encoding WT protein or the indicated mutations.

Same explanation goes for figure 5.

Minor concerns

10. Mammalian Sting signaling is discussed in the manuscript, but it should also be mentioned if the Sting, Fadd and caspase interaction surfaces are conserved in mammals and whether they are involved in NF- κ B signaling induced via mammalian Sting.

We have already discussed this. Unfortunately, the literature is currently quite slim and not fully consistent on this topic. We will add one sentence to the discussion to clarify that the residues involved in the interaction with dFADD are not conserved in vertebrate STING.

Citing from the discussion: Newly added text in italic.

Interestingly, a role for mammalian FADD in STING signaling was suggested in the first report identifying STING as a key regulator of innate immune signaling in mammals (Ishikawa and Barber, 2008), although little follow up is available in the literature. Intriguingly, a role for FADD/Caspase 8 in antiviral immunity and IFN induction was proposed as early as 2004 (Balachandran et al., 2004). Yet, we find that the amino acids in dSTING, which are involved in the dFADD interaction, are only conserved among invertebrate STING and not vertebrate STING. Subsequent studies confirmed a role of FADD and Caspase 8 in inflammatory signaling, including response to the synthetic dsRNA analogue poly(I:C), although the underlying mechanisms are still poorly understood and the response to typical RIG-I activating viruses remained largely intact in FADD deficient cells (Kawai et al., 2005, Seth et al., 2005, Yoneyama et al., 2005, Takahashi et al., 2006, Balachandran et al., 2007).

11. The rationale behind using injection of 2'3'-cGAMP/3'2'-cGAMP to activate the Sting pathway to protect from DCV infection, which should induce the same pathway is not explained.

Please see response to R2, minor point 2.

12. In the manuscript, it is written that the response of Tak1^{-/-} flies to Gram negative bacteria (ED 4e) was highly impaired although the fold induction of AttA is significantly induced (with ****). It would be more correct to say that it is reduced compared to control.

The text has been revised accordingly.

13. In figure 2 and 3, in the figure markings and figure legends all transfected plasmids are not indicated (Relish-V5 and Sting-HA).

This has been corrected.

14. The correct nomenclature for fly proteins and genes should be used (as described in FlyBase).

We have done this.

Referee #2:

The report by Winther et al. addresses the molecular mechanisms related to the activation of cGAS/STING signalling in *Drosophila melanogaster* and, in particular, how STING activates the NF- κ B family transcription factor, Relish. The manuscript details the authors' efforts in delineating the steps required for Relish activation downstream of crucial antiviral effectors involved in STING signalling. The authors use a combination of in vitro CRISPR efforts in *Drosophila* S2 cells combined with in vivo experiments in flies to suggest that FADD and DREDD are crucial for STING-dependent induction of Relish cleavage and downstream expression of STING target genes. The authors also use structural biology predictions to suggest that FADD directly binds to STING oligomers and that this binding is important for the response to viral infection. Moreover, again using structural biology predictions, the authors describe the potential molecular mechanisms by which FADD interacts with DREDD and IMD and provide a potential explanation for the fact that FADD and DREDD, but not IMD, are involved in both STING antiviral signalling and IMD anti-bacterial signalling.

In general, experiments and data are well presented and appropriately controlled and the data is of high quality. The authors provide extensive extended data to complement the main figures and make a compelling case for their proposed model of how STING signalling is regulated. The manuscript should be of great interest for researchers working on mechanisms regulating the response to viral infection and for the larger innate immunity field.

Comments on the manuscript are shown below.

Major points:

1. Regarding Extended Data Figure 1, authors should comment on why the Relish KO cells appear to have a higher PCR amplicon than the control WT cells? Do the cells have any duplication of sequences to account for this observation?

In the *Relish* KO cell an insertion happened, in reality several different insertions occurred (cells have multiple chromosomes). We verified this by sequencing.

2. Authors should comment on why the non-cleavable Relish mutant (D545A) is acting as a dominant negative allele. Also, authors should comment on why the C-terminally truncated form of Relish (1-545) seems to be much more efficient in activating the STING reporter than the IMD reporter at basal levels without stimulation (Extended Data Figure 2a,c).

We speculate that ectopic expression of non-cleavable Relish accumulates and competes with the endogenous protein for binding to the dFADD-DREDD complex, and since it is non-cleavable, it will be a competitive inhibitor.

We do not have a molecular explanation at this stage for the high activity of Relish (1-545) on the dSTING vs. IMD reporter, but we hypothesize that it could be related to association of cofactors explaining the specificity of the transcriptional response downstream of dSTING or IMD (e.g., limiting amounts of DIF, previously shown to heterodimerize with Relish to regulate AMP genes), may limit activity of Relish on the IMD reporter^{11,12}.

3. One important point that the authors do not explore is whether there is any synergy, competition or cross-activation between the STING and IMD pathways. In nature, it is conceivable that injury would result in simultaneous infection with bacteria and viruses. How would this be tackled by the immune system? Does PGRP-LC lead to activation of the STING reporter? Does cGLR1/2 activate the IMD reporter? Is there any indication that there is a crosstalk between the two pathways? The fact that both share components could indicate that this is a possibility.

This is indeed an interesting question. We reported earlier that activation of the dSTING pathway results in upregulation of the antimicrobial peptides *in vivo* (see Fig. 2c in¹³). We carefully analyzed the cross activation of the two reporters when establishing our experimental system and our data showed that, while cGLR1 did not activate the IMD reporter, PGRP-LC could result in limited activation of the dSTING reporter, see figure below. But in general, the cross reactivity is less than what would be expected given that the two pathways share Relish as the key transcription factor, which is also highlighted by the strong induction of the dSTING but not the IMD reporter by expressing the pre-cleaved form of Relish [1-545]. Yet, some degree of crosstalk *in vivo* is likely, and we have added a sentence in the discussion to acknowledge this.

4. Authors should discuss why 3'2'-cGAMP was used in the experiments with the Tak1 mutant and not in others. Is there a chance that some phenotypes are dampened due to the use of 2'3'-cGAMP?

We apologize for this omission. In fact, the initial experiments on Relish (Fig. 1b and c ; Extended Data Fig. 3) were performed before we knew of 3'2'-cGAMP. Nevertheless, cGLR1 produces mostly 3'2'-cGAMP in S2 cells¹⁴, so the data presented in Fig.1a-c are coherent. In addition, analysis of our published RNA-Seq data of flies stimulated with 2'3'-cGAMP¹³ or 3'2'-cGAMP¹⁵ reveal that the response to 3'2'-cGAMP is stronger but that the set of genes upregulated is similar to 2'3'-cGAMP. We have added a sentence in Methods and the Results section to acknowledge this point and avoid confusion.

5. Authors should clarify or test whether STING overexpression is sufficient to activate the STING reporter in dFADD KO cells. This would strengthen the argument that dFADD is essential for STING signalling.

We have clearly shown that dSTING signalling in response to cGLR1 transfection is lost in *Fadd* KO cells AND that the respond to cGAMP injection is lost *in vivo* in *Fadd* deficient flies. We overexpressed dSTING as part of the rescue experiments in Fig. 1f and this only led to a marginal signal and requires co-expression with cGAS to active full signalling, which is expected as this system is quite tightly controlled as unwanted activation carries a significant fitness cost.

6. dFADD re-expression in dFADD KO cells is not shown in WB experiments.

We believe there is a misunderstanding here, the data are shown in Extended Data 7, panel A.

7. Authors should comment on why the baseline of Srg gene expression is so low in FADD mutants.

The reviewer is correct; there is indeed a significant decrease in the basal level of Srgs in the *Fadd* mutant. We note that the basal expression level of some Srgs (e.g., Srg3) is also significantly decreased in *Relish* and *Dredd* mutant flies. Furthermore, we also see reduced background signalling in our KO cells. We believe there is a certain homeostatic level of dSTING signalling, which can either be maintained through activation of the cGLRs by endogenous ligands or in the case of flies, by signals originating from the microbiota. As key proteins in the pathway, like dSTING, are driven by dSTING signalling, disruption of the pathway lowers the homeostatic levels of Srgs

8. Authors should test whether the dFADD mutations created to abrogate binding to STING affect IMD signalling. If so, this could reveal structural changes in dFADD beyond simply affecting interaction with STING.

We have actually performed this experiment, and the results are presented in Extended Data Fig. 11b. We also discuss it quite detailed in the text. As it is the same area of dFADD that binds both dSTING and IMD, it is no surprise that there is some overlap, but since those mutations do not affect the binding to DREDD, we do not believe there are major structural rearrangements. Citing end of page 17 of our manuscript.

“ However, the same was not true for R164A/D181A and R164A/D181A/E184A mutations of dFADD which were designed to abolish dSTING activation. The double mutation R164A/D181A almost abolished IMD signaling, whereas the triple mutation R164A/D181A/E184A had a more intermediate phenotype (Extended Data Fig. 11b). Since it is the same area of dFADD which is responsible for interacting with either dSTING or IMD, it is not surprising that the mutations designed to abolish the binding to dSTING also have some effect on the interaction with IMD. We cannot offer an explanation to why the triple mutation is less severe than the double mutation. As controls, we also tested if the substitutions R171A and E177A affected the ability of dFADD to bind DREDD, which they did not (Extended Data Fig. 11c).”

9. Related to the previous point, authors should comment on whether the E183A and R240A

mutations in STING affect STING dimerization. Additionally, authors could test whether dFADD can co-IP a STING mutant that cannot dimerise. According to the authors' proposed model, dFADD should not meaningfully interact with a monomeric STING molecule.

This is a valid question, yet the same technical limitations as discussed earlier (R1, point1), which prevent us from measuring the oligomeric state of STING, also prevent a meaningful answer to this.

10. If possible, authors should provide evidence that the dIKK β -dIKK γ complex is not required for STING-induced Relish cleavage and downstream induction of STING target genes to complement the analysis of the dTak1-dTab2 complex.

This is an interesting important question that will be the topic of a future report. We did show that at least dIKK β is involved in dSTING signalling in our initial report by Goto et al 2018¹⁶. This is also clearly acknowledged in the introduction.

“Indeed, our initial characterization of the dSTING pathway only pointed to a contribution of the dIKK β kinase”

11. According to Ertürk-Hasdemir et al. (2009), it should be possible to detect phosphorylated Relish in the S2 system (S528/S529). Is Relish phosphorylation similarly affected by activation of the IMD pathway or the STING pathway? Based on the authors' observations and the lack of effect of dTak1, one would potentially predict that although both lead to Relish cleavage, activation of STING signalling may not lead to Relish phosphorylation (or at least not to the same extent). Authors should therefore discuss the importance/relevance of Relish phosphorylation for Relish cleavage in the context of STING signalling.

This is an interesting and important question that we are actively seeking an answer to. First of all, we would like to remind the reviewer that our initial report by Goto et al from 2018, clearly showed that dIKK β is needed for dSTING signalling¹⁶. We are currently generating and characterising CRISPR-based KO of both dIKK β and dIKK γ in S2 cells, so that we can address this question in detail. However, this is a full project in itself and will be the topic of a future report by us.

12. Authors should investigate downstream Relish cleavage in S2 cells when the dFADD mutations

R164A, D181A and E184A are present (alone or in combination). Similarly, the effect of the mutations that affect the dFADD-DREDD interaction (dFADD R39A/R40A and DREDD E131A/E167A) on Relish cleavage should also be monitored.

Please see answer to R1, Q5

13. Authors should comment on whether the dFADD-STING interaction is compatible with the dFADD-DREDD interaction or whether they are mutually exclusive as that has relevance for their proposed model.

dFADD is composed of two separate death fold domains (a death domain (DD) and a death effector domain (DED)), connected by a linker. dFADD binds to dSTING via its DD and to DREDD via its DED, thus the two interactions are clearly independent, as is also clearly shown in our final model presented in Fig. 6.

14. Regarding Extended Data Figure 10d, there is a potential issue with the prediction of dFADD-IMD interaction as most of the IMD molecule is a very low confidence prediction. How could this influence the proposed mechanism?

This part of the IMD protein is intrinsically disordered, as is the case for many signalling proteins. However, the DD of IMD has a high confidence and this is where the predicted interaction occurs. Due to the modular nature of protein folds, disordered parts of a protein do not have a negative effect on other folded parts. Thus, it does not affect the prediction. Most people remove disordered part from the final figures that they show in papers, we believe it is clearer to show the full protein.

Minor points:

Related to Figure 3a, there is a mistake in the nomenclature of the dSTING mutation and aa residue. Panel 3a mentions D183 but panel 3c and the main text mentions E183. This should be corrected in the final version of the manuscript.

We apologize for the mistake, which has been corrected.

Related to Figure 1, authors should clarify which PGRP-LC isoform(s) are being expressed in S2 cells.

We have clarified that we use a vector expressing PGRP-LCa as described in ¹⁷, reference is also added in M&M section.

Authors observe that RelishD545A mutant flies are more susceptible to DCV infection. Is this also the case if flies are raised in axenic conditions?

We have not performed this experiment, but as mentioned above, the question of the microbiota and the cross-talk between the dSTING and IMD pathway are now discussed.

Protection by 2'-3'-cGAMP injection does not appear to be very extensive even when flies are WT. Why is this? Does expression of cGLR1 lead to a similar level of rescue or a more effective rescue? Are there other, more important antiviral pathways at play?

Indeed, expression of cGLR1 triggers a stronger protection than cGAMP injection, and we hypothesize that this is due to the sustained production of CDNs in the transgenic flies as we now know that the concentration of injected CDNs rapidly decreases in flies. In addition, we also showed that cGLRs produce several CDNs *in vivo*¹⁵. In WT flies there is still a protection arising from the endogenous cGLRs producing CDNs, which lowers WT flies sensitivity to viral infection. Finally, all these experiments are performed in flies with an intact RNA interference pathway, which is a major barrier to viral infections in flies.

Related to Figure 3f,g and the co-localisation of dFADD and STING in S2 cells, authors should perhaps increase the number of cells analysed given that dFADD and STING appear to have large areas where they do not colocalise.

Please see respond to R1. Q3.

The inclusion in the Introduction of a discussion on the production of CDNs in bacteria is perhaps not necessary to provide a relevant background for the experimental work performed.

We believe that the comments of the reviewers on the use of axenic flies and the possible cross-talk between antibacterial and antiviral immunity in flies, argue to keep these two sentences, but we are flexible.

The sentences in question are: *“In bacteria, cGAS-like enzymes are activated upon sensing of phage infection and trigger the production of cyclic di- or tri-nucleotides (CDNs, CTNs), which bind to and activate effector antiphage molecules. These effectors include, although they are not limited to, STING homologues^{2,7}.”*

In most of the Extended Data Figures, the expressed constructs are not detectable in WB when at low concentration and, therefore, do not really serve as controls.

The rescue experiment requires the use of low amount of plasmid in order not to overload the system in cells, however, this low level of protein expression is not detectable by western blotting, partly due to the low transfection efficiency of S2 cells. Therefore, it was important to include transfection with higher concentrations to prove that the vectors we constructed are functional. We agree it is a technical limitation but believe it is the best possible solution.

Histone H3 loading control blots could have been optimised in some of the figure panels as they seem overexposed making it more challenging to clearly determine equal protein loading in all samples.

We apologize for the poor quality of histone H3 blots, resulting from the fact that we were using the same membranes to monitor the different proteins. In addition, we want to stress that (i) we could not use actin as a control because of its size overlapping those of the proteins we were monitoring ; (ii) although it was difficult to get a sharper signal because of the abundance of H3, the signals were not saturated in the Chemidoc used for detection ; (iii) the background visible for the other antibodies was coherent with equilibrated loading.

Methods section is thorough, with only minor omissions (cell number used in WB experiments, transfection reagent used in dSTING IP and MS experiments).

We apologise and will correct this.

Minor typos present (particularly in Methods section).

We apologise and have corrected this

Referee #3:

Referee Report on EMBO J Submission: "FADD is recruited to activated STING oligomers initiating caspase-mediated NF- κ B activation in *Drosophila melanogaster*"

This is an interesting and timely study that addresses an important mechanistic question in the field of STING signaling. The authors provide convincing evidence that FADD is recruited to oligomerized STING complexes in *Drosophila* and functions downstream to activate the NF- κ B-like transcription factor Relish. The structural modeling of FADD interaction with STING oligomers, together with functional data, advances our understanding of how the conserved FADD/caspase axis is integrated into distinct immune pathways. The work is carefully executed, the conclusions are significant, and the manuscript is clearly written. Overall, I find the study appropriate for publication in The EMBO Journal.

That said, I have one major suggestion that I believe would substantially strengthen the manuscript. The central mechanistic claim is that FADD is recruited to STING via a specific interface that the authors have modeled in detail. While the biochemical and structural analyses are compelling, the model would be much more convincing if supported by genetic data *in vivo*. In particular, the authors could generate dFADD variants defective in STING binding (as defined by their structural predictions) and introduce these variants into the dFADD locus. Testing such flies for (i) susceptibility to viral infection and (ii) partial protection upon 3'2'-cGAMP injection (analogous to the experiments presented in Fig. 1c with Relish D545A flies) would provide powerful validation of the model. Even partial data in this direction would significantly strengthen the manuscript. Apart from this, I have no major concerns.

We thank the reviewer for the positive comments. While we agree that introducing point mutations in dFADD and testing the phenotype of the knock-in flies would reinforce the conclusion of the manuscript, we want to stress that it would require 6 to 8 months to obtain the mutant flies and carry considerable costs. Given that we have already provided extensive biochemical characterisation of this interaction, we do not believe that the eventual gain by this experiment outweigh the significant delay of publication of our findings and the associated costs.

Minor points could be addressed during revision, the authors may wish to expand briefly in the discussion on how the FADD-STING interaction compares to vertebrate STING signaling, which could help highlight the evolutionary insights.

We agree, please see R1. Point 10.

- 1 Shang, G., Zhang, C., Chen, Z. J., Bai, X. C. & Zhang, X. Cryo-EM structures of STING reveal its mechanism of activation by cyclic GMP-AMP. *Nature* **567**, 389-393 (2019).
<https://doi.org:10.1038/s41586-019-0998-5>
- 2 Ergun, S. L., Fernandez, D., Weiss, T. M. & Li, L. STING Polymer Structure Reveals Mechanisms for Activation, Hyperactivation, and Inhibition. *Cell* **178**, 290-301 e210 (2019).
<https://doi.org:10.1016/j.cell.2019.05.036>
- 3 Kim, C. H., Paik, D., Rus, F. & Silverman, N. The caspase-8 homolog Dredd cleaves Imd and Relish but is not inhibited by p35. *J Biol Chem* **289**, 20092-20101 (2014).
<https://doi.org:10.1074/jbc.M113.544841>
- 4 Ishikawa, H. & Barber, G. N. STING is an endoplasmic reticulum adaptor that facilitates innate immune signalling. *Nature* **455**, 674-678 (2008).
<https://doi.org:10.1038/nature07317>
- 5 Balachandran, S., Thomas, E. & Barber, G. N. A FADD-dependent innate immune mechanism in mammalian cells. *Nature* **432**, 401-405 (2004).
<https://doi.org:10.1038/nature03124>
- 6 Kawai, T. *et al.* IPS-1, an adaptor triggering RIG-I- and Mda5-mediated type I interferon induction. *Nat Immunol* **6**, 981-988 (2005). <https://doi.org:10.1038/ni1243>
- 7 Seth, R. B., Sun, L., Ea, C. K. & Chen, Z. J. Identification and characterization of MAVS, a mitochondrial antiviral signaling protein that activates NF-kappaB and IRF 3. *Cell* **122**, 669-682 (2005). <https://doi.org:10.1016/j.cell.2005.08.012>
- 8 Yoneyama, M. *et al.* Shared and unique functions of the DExD/H-box helicases RIG-I, MDA5, and LGP2 in antiviral innate immunity. *Journal of immunology* **175**, 2851-2858 (2005).
<https://doi.org:10.4049/jimmunol.175.5.2851>
- 9 Takahashi, K. *et al.* Roles of caspase-8 and caspase-10 in innate immune responses to double-stranded RNA. *Journal of immunology* **176**, 4520-4524 (2006).
<https://doi.org:10.4049/jimmunol.176.8.4520>
- 10 Balachandran, S., Venkataraman, T., Fisher, P. B. & Barber, G. N. Fas-associated death domain-containing protein-mediated antiviral innate immune signaling involves the regulation of Irf7. *Journal of immunology* **178**, 2429-2439 (2007).
<https://doi.org:10.4049/jimmunol.178.4.2429>
- 11 Morris, O. *et al.* Signal Integration by the I kappa B Protein Pickle Shapes Drosophila Innate Host Defense. *Cell Host Microbe* **20**, 283-295 (2016).
<https://doi.org:10.1016/j.chom.2016.08.003>
- 12 Tanji, T., Yun, E. Y. & Ip, Y. T. Heterodimers of NF-kappaB transcription factors DIF and Relish regulate antimicrobial peptide genes in Drosophila. *Proc Natl Acad Sci U S A* **107**, 14715-14720 (2010). <https://doi.org:10.1073/pnas.1009473107>

- 13 Cai, H. *et al.* 2'3'-cGAMP triggers a STING- and NF-kappaB-dependent broad antiviral response in *Drosophila*. *Sci Signal* **13** (2020). <https://doi.org:10.1126/scisignal.abc4537>
- 14 Slavik, K. M. *et al.* cGAS-like receptors sense RNA and control 3'2'-cGAMP signaling in *Drosophila*. *Nature* (2021). <https://doi.org:10.1038/s41586-021-03743-5>
- 15 Cai, H. *et al.* The virus-induced cyclic dinucleotide 2'3'-c-di-GMP mediates STING-dependent antiviral immunity in *Drosophila*. *Immunity* **56**, 1991-2005 e1999 (2023). <https://doi.org:10.1016/j.immuni.2023.08.006>
- 16 Goto, A. *et al.* The Kinase IKKbeta Regulates a STING- and NF-kappaB-Dependent Antiviral Response Pathway in *Drosophila*. *Immunity* **49**, 225-234 e224 (2018). <https://doi.org:10.1016/j.immuni.2018.07.013>
- 17 Bonnay, F. *et al.* Akirin specifies NF-kappaB selectivity of *Drosophila* innate immune response via chromatin remodeling. *The EMBO journal* **33**, 2349-2362 (2014). <https://doi.org:10.15252/emj.201488456>

Dear Rune,

Thank you again for the submission of your revised manuscript (EMBOJ-2025-121985R) to The EMBO Journal for our consideration, and for your patience during peer review. As I have already informed you, the three original referees, who had also assessed the previous version of your manuscript, have seen your revision, and we have received their comments, which I have already shared with you (they are appended again below).

I am pleased to say that the referees recognize the high quality of the work, which they find interesting and timely. Reviewer #1 has no further comments, whereas referees #2 and #3 point out that most of the changes in the revised manuscript are textual, some of the initially raised concerns have not been addressed, and some of the data that were included in your rebuttal letter could be incorporated in the manuscript to strengthen it further.

On balance, and taking into consideration the scope and bar of The EMBO Journal, I would like to invite you to submit a final version of your manuscript as soon as possible. As we have previously discussed, no more far-reaching experiments will be required for the publication of this manuscript here. I would only kindly ask you to consider the suggestion of referee #2 regarding moving some of the data of your point-by-point response to the manuscript (as Expanded View or Appendix Figures), as we discussed earlier. Please include in your resubmission a brief point-by-point response detailing any changes in the manuscript or reorganization of data in new Figures.

From the editorial side, there are also a few changes we need you to make in the final version of your manuscript, before we can move forward with its formal acceptance and publication in The EMBO Journal:

- Please make sure that all deposited data to external repositories are made publicly available at the time of publication and listed in the Data Availability section of the revised manuscript. For each dataset, please provide the database, dataset identifier, and specific and permanent URL. The referee access information is no longer necessary and can be removed.
- Please change heading "Declaration of interest" to "Disclosure and competing interests statement".
- Please provide the e-mail addresses of the co-corresponding authors on the title page of the manuscript.
- Please note that the reference format is not correct. The heading of the list should be "References"; the names of the first 10 co-authors of each publication should be listed, followed by "et al.". Please see our Reference Guidelines for more information: <https://link.springer.com/journal/44318/submission-guidelines#cms-Reference-guidelines>.
- Please note that the funding information provided in the Acknowledgements section of the manuscript should match that provided in the online system; currently, the following funders and grants are acknowledged in the manuscript but missing in the system: Hoffmann Infinitus Program, 2022A0505030018, PhD fellowships from MENRT and the Fondation pour la Recherche Médicale and the Fondation ARC.
- The Reagents and Tools Table should be uploaded separately as a "Reagent Table" file. Please remove this Table from the main manuscript file.
- Please consider incorporating the list of oligos currently found in the Appendix file in the Reagents and Tools Table.
- Similarly, the "sgRNA target sites" from your Methods section could be included in the Reagents and Tools Table.
- Please note that EMBO press papers are accompanied online by:
 - A) a short (2 sentences) summary of the findings and their significance,
 - B) 2-5 short bullet points highlighting the key results, and
 - C) a synopsis image in .jpg or .png format that is exactly 550 pixels wide and 300-600 pixels high (the height is variable). Please note that all text needs to be legible at the final size.Please upload this information along with your revised manuscript (the text for A and B should be provided in a separate Word file).
- During our standard pre-publication Figure integrity checks, our team noticed that the EV Figures of your manuscript appear rather pixelated. We would thus kindly request you upload the source data for blots included in your EV Figures (please combine them in a single ZIP folder).
- Our data editors have checked your Figures and their legends, and raised the following queries. Please address all points below completely in your revised manuscript (all changes should be highlighted or "tracked"):
 - Please note that the legend for Figure EV1 is not provided in the sequential manner. This needs to be rectified.
 - Please provide the exact p-values for Figures 1B, C, D, E, F, G; 2B, C, E, F; 3B-D; 4B, C, E, F; 5B, D, F; EV1 A, C, E, F, G, H;

EV2 B-G; EV3 B-G.

- Please note that information related to "n" is missing in the legend of Figure 4F.

- Please note that the error bars are not defined in the legend of Figures EV5 B.

- The manuscript sections need to be named and ordered as follows: Title page - Abstract - Introduction - Results - Discussion - Methods - Data Availability - Acknowledgements - Disclosure and Competing Interests Statement - References - Figure Legends - main Tables (if there are any) - Expanded View Figure Legends.

- Please also note that as part of the EMBO Press transparent editorial process, The EMBO Journal publishes online a Peer Review File along with each accepted manuscript. This File will be published in conjunction with your paper and will include the referee reports, your point-by-point responses and all pertinent correspondence relating to the manuscript. Your Author's Checklist will also be published at the end of the Peer Review File. Please let us know in case you want to remove any data or figures from your point-by-point responses before they are published as part of the Peer Review File. Retaining unpublished data in the Peer Review File means that these count as published and that the Peer Review File would need to be referenced in future publications. Please let the editorial office know in case you want to remove any data from this file (contact@embojournal.org).

We look forward to seeing a final version of your manuscript as soon as possible. Please let us know if you have any questions and use this link to submit your revision: <https://emboj.msubmit.net/cgi-bin/main.plex>.

Best regards,

Ioannis

Referee #1:

The authors have addressed all my comments and I do not have any further concerns.

Referee #2:

The report by Winther et al. addresses the molecular mechanisms related to the activation of cGAS/STING signalling in *Drosophila melanogaster*. The report is particularly focused on how STING activates Relish and features the authors' efforts in delineating the steps required for Relish activation downstream of STING-related antiviral effectors. The authors use complementary approaches combining in vitro CRISPR efforts in S2 cells with in vivo experiments to suggest that FADD and DREDD are crucial for STING-dependent induction of Relish cleavage and downstream expression of STING target genes. The authors also present structural biology predictions to suggest that FADD directly binds to STING oligomers and that this binding is important for the response to viral infection. In addition, the authors describe potential molecular mechanisms by which FADD interacts with DREDD and IMD and potential explanations for the fact that FADD and DREDD, but not IMD, are involved in both STING antiviral signalling and IMD anti-bacterial signalling, supported by in silico structural biology predictions.

In general, experiments and data are well presented and appropriately controlled and the data is of high quality. The authors provide extensive extended data to complement the main figures and make a compelling case for their proposed model of how STING signalling is regulated. The manuscript should be of great interest for researchers working on mechanisms regulating the response to viral infection and for the larger innate immunity field.

The revised manuscript is, disappointingly, not much different from the original submission, despite the fact that all reviewers raised pertinent points to improve the quality of the manuscript. Changes implemented are mostly relating to the text. The authors provide answers in their rebuttal letter for why certain approaches were not undertaken or their results not included in the final version of the manuscript, including the fact that the first author is no longer hosted by the research laboratory. Given the high quality of the original submission, this is a compelling report that will advance the field and contribute to enhancing our understanding of how immunity is regulated in flies and beyond. However, authors should consider if including some of the data provided in the rebuttal letter will complement and extend the data currently included in the manuscript.

Referee #3:

The authors did not address my unique major comments with even partial data as indicated in my review... Hence I leave it to the editor to decide if the resolution of the work is sufficient for EMBO J. . That said and as initially said in my first review the work is of quality, interesting and timely.

Response to referees.

Referee #1:

The authors have addressed all my comments and I do not have any further concerns.

Much appreciated.

Referee #2:

The report by Winther et al. addresses the molecular mechanisms related to the activation of cGAS/STING signalling in *Drosophila melanogaster*. The report is particularly focused on how STING activates Relish and features the authors' efforts in delineating the steps required for Relish activation downstream of STING-related antiviral effectors. The authors use complementary approaches combining in vitro CRISPR efforts in S2 cells with in vivo experiments to suggest that FADD and DREDD are crucial for STING-dependent induction of Relish cleavage and downstream expression of STING target genes. The authors also present structural biology predictions to suggest that FADD directly binds to STING oligomers and that this binding is important for the response to viral infection. In addition, the authors describe potential molecular mechanisms by which FADD interacts with DREDD and IMD and potential explanations for the fact that FADD and DREDD, but not IMD, are involved in both STING antiviral signalling and IMD anti-bacterial signalling, supported by in silico structural biology predictions.

In general, experiments and data are well presented and appropriately controlled and the data is of high quality. The authors provide extensive extended data to complement the main figures and make a compelling case for their proposed model of how STING signalling is regulated. The manuscript should be of great interest for researchers working on mechanisms regulating the response to viral infection and for the larger innate immunity field.

We appreciate the praise!

The revised manuscript is, disappointingly, not much different from the original submission, despite the fact that all reviewers raised pertinent points to improve the quality of the manuscript. Changes implemented are mostly relating to the text. The authors provide answers in their rebuttal letter for why certain approaches were not undertaken or their results not included in the final version of the manuscript, including the fact that the first author is no longer hosted by the research laboratory. Given the high quality of the original submission, this is a compelling report that will advance the field and contribute to enhancing our understanding of how immunity is regulated in flies and beyond. However, authors should consider if including some of the data provided in the rebuttal letter will complement and extend the data currently included in the manuscript.

The data showing that IMD mutations do not affect the STING reporter and vice versa has been moved from the cover letter to Appendix figure 5.

As suggested, we added two panels to EV Fig 4 f and g, showing abundance of STING and DREDD peptides respectively, in the Mass spectrometric data from STING IP experiment shown in figure (Fig 3d). This verifies that the abundance of STING peptides do not change after activation by cGAS, but DREDD peptides are detected after cGAS activation of STING, confirming that DREDD is recruited to the activated STING signalosome.

Referee #3:

The authors did not address my unique major comments with even partial data as indicated in my review... Hence I leave it to the editor to decide if the resolution of the work is sufficient for EMBO J. . That said and as initially said in my first review the work is of quality, interesting and timely.

While the suggested experiments (the generation of knock in flies of some of the key mutations described in this paper and the corresponding in vivo confirmation of our data from S2 cells) are scientifically valid, this would delay our paper to an extent where it would no longer be timely. We also believe, as described in our previous rebuttal, that those experiments are not essential to the mechanistical argument we are making.

Dear Rune,

Congratulations on an excellent manuscript! I am very pleased to inform you that it has been accepted for publication in The EMBO Journal. Thank you for addressing the initially raised referee concerns and our editorial requests for corrections and changes.

You may qualify for financial assistance for your publication charges - either via a Springer Nature fully open access agreement or an EMBO initiative. Check your eligibility: <https://link.springer.com/journal/44318/how-to-publish-with-us>

If you have any questions, please do not hesitate to contact the Editorial Office. Thank you for your contribution to The EMBO Journal. Working with you has been a pleasure.

Best regards,

Ioannis

Please note that it is The EMBO Journal policy for the transcript of the editorial process (containing referee reports and your response letters) to be published as an online supplement to each paper. If you should prefer removal of any referee-only figures included in the point-by-point response(s), e.g. because they may still be used for future publication or because they have been reproduced from published work by others, please do let us know immediately via response email.

More information is available here: <https://link.springer.com/partners/embo-press/editorial-policies#Peer%20review>
